# Towards a Golden Classifier-Free Guidance Path via Foresight Fixed Point Iterations

**Kaibo Wang[1], Jianda Mao[1], Tong Wu[1], Yang Xiang[1,2]***

[1]Department of Mathematics, The Hong Kong University of Science and Technology
[2] Shenzhen-Hong Kong Collaborative Innovation Research Institute, HKUST
{kwangbi, jmaoao, twubi}@connect.ust.hk, maxiang@ust.hk

## Abstract

Classifier-Free Guidance (CFG) is an essential component of text-to-image diffusion models, and understanding and advancing its operational mechanisms remains a central focus of research. Existing approaches stem from divergent theoretical interpretations, thereby limiting the design space and obscuring key design choices. To address this, we propose a unified perspective that reframes conditional guidance as fixed point iterations, seeking to identify a *golden path* where latents produce consistent outputs under both conditional and unconditional generation. We demonstrate that CFG and its variants constitute a special case of single-step short-interval iteration, which is theoretically proven to exhibit inefficiency. To this end, we introduce Foresight Guidance (FSG), which prioritizes solving longer-interval subproblems in early diffusion stages with increased iterations. Extensive experiments across diverse datasets and model architectures validate the superiority of FSG over state-of-the-art methods in both image quality and computational efficiency. Our work offers novel perspectives for conditional guidance and unlocks the potential of adaptive design.[1]

## 1 Introduction

Diffusion models [14, 27, 16] have emerged as a transformative paradigm for conditional generation, achieving remarkable success in synthesizing high-fidelity images from text prompts [24, 22]. Classifier-free guidance (CFG) [15] serves as a key component, steering the generation towards prompt alignment by amplifying the conditional outputs of a model. However, its over-amplified guidance introduces critical trade-offs, including compromised image quality and limited diversity [5], which highlight the necessity for deeper theoretical and practical insights into guidance mechanisms.

Existing efforts to mitigate the limitations of CFG are typically based on distinct *conditional sampling* perspectives. For example, some methods conceptualize CFG as sampling from sharpened probability distributions [8, 9], while others mitigate off-manifold deviations through posterior sampling [5] or improve semantic alignment through reflective sampling [1, 20]. Although these approaches are theoretically sound, the absence of a unified interpretation narrows the design space of guidance mechanisms, rendering each method a tightly coupled framework where components cannot be independently modified. To integrate the effective components of existing approaches and further unlock the potential of guidance mechanisms, a critical question arises:

> *How can we systematically explore the design space for conditional guidance and develop more effective algorithms from a unified perspective?*

---

*Corresponding author

[1]The codes are available at https://github.com/Ka1b0/Foresight-Guidance.

39th Conference on Neural Information Processing Systems (NeurIPS 2025).

We address this question by proposing a unified framework based on fixed point iterations. Observing that the latent variable $x_t$ achieves better generation quality and alignment when its unconditional generation matches the prompt-conditioned generation [1, 34] (Figure 1 (a)), we reframe conditional generation as identifying a *golden path* composed of such points. Our framework comprises two decoupled steps: calibration and denoising. At each timestep $t$, the *calibration step* iteratively refines $x_t$ to $\hat{x}_t$ via fixed point iterations. Following this, the *denoising step* samples from $p(x_{t-1} \mid \hat{x}_t)$ with unconditional noise prediction, as depicted in Figure 1 (b). We demonstrate that CFG [15] and its variants [5, 1, 20] are special cases of this framework. By disentangling guidance from sampling, our approach enables (1) systematic comparison of design choices across methods and (2) upgrades existing algorithms through fixed point iteration design.

The number of iterations and the consistency intervals in fixed point algorithms are critical yet underexplored design dimensions. Current methods typically solve multiple short-interval subproblems using a single iteration each, a suboptimal strategy proven by our theoretical analysis. We first demonstrate that existing methods can be upgraded simply by increasing the iteration count (e.g., CFG $\times K$). Beyond that, we propose *Foresight Guidance* (FSG), which prioritizes solving longer-interval subproblems during the earlier stages of the diffusion process with more iterations (Figure 1 (c)). FSG enhances alignment by propagating guidance signals over extended time horizons while improving efficiency by reducing the number of subproblems.

Extensive experiments across datasets and models demonstrate that existing guidance methods directly benefit from increased iterations, and FSG further improves generation quality with lower computational overhead. We provide novel insights into advancing the development of efficient and adaptable guidance mechanisms. Our contributions are threefold:

- We model conditional guidance as a calibration task toward a golden path, unifying CFG and related methods under a fixed point iteration framework.
- We upgrade existing methods and propose FSG by designing consistency intervals and iteration schedules, achieving better alignment and efficiency.
- We validate the effectiveness of FSG through comprehensive experiments and demonstrate the potential of the framework in guidance design.

## 2 Preliminaries

### 2.1 Diffusion Models

Diffusion models [14, 27] are generative models that synthesize images by progressively reversing a predefined noising process. The forward diffusion process ($t = 0 \rightarrow T$) gradually corrupts an initial clean image $x_0$ through a noising function $n_{t \rightarrow t+1}$ determined by a noise schedule $\alpha_{1:T}$:

$$x_{t+1} = n_{t \rightarrow t+1}(x_t) = \sqrt{\alpha_{t+1}}x_t + \sqrt{1 - \alpha_{t+1}}\epsilon, \quad \epsilon \sim \mathcal{N}(0, \mathbf{I}). \tag{1}$$

The reverse process reconstructs the data by iteratively denoising $x_t$ to $x_{t-1} \sim p(x_{t-1} \mid x_t)$ through $f_{t \rightarrow t-1}$, which consists of a neural network-based noise predictor $\epsilon(x_t)$ and a reverse sampler:

$$x_{t-1} = f_{t \rightarrow t-1}(x_t) = \text{Sampler}(x_t, \epsilon(x_t)), \tag{2}$$

where the sampler design is flexible, provided that $x_t$ follows the marginal distribution $p_t(x_t)$ of the forward process. Notably, DDIM [27] operates as a deterministic sampler, corresponding to a discrete ODE formulation [28]. Let $\bar{\alpha}_t = \prod_{s=1}^{t} \alpha_s$, its update rule is defined as:

$$\phi(x_t) = \frac{x_t - \sqrt{1 - \bar{\alpha}_t}\epsilon(x_t)}{\sqrt{\bar{\alpha}_t}},$$
$$x_{t-1} = \sqrt{\bar{\alpha}_{t-1}}\phi(x_t) + \sqrt{1 - \bar{\alpha}_{t-1}}\epsilon(x_t). \tag{3}$$

To generate an image, we first sample $x_T \sim \mathcal{N}(0, \mathbf{I})$ and then compute $x_0 = f_{T \rightarrow 0}(x_T) = f_{1 \rightarrow 0} \circ \cdots \circ f_{T \rightarrow T-1}(x_T)$.

### 2.2 Classifier-Free Guidance

In conditional generation tasks (e.g., generating images for a given prompt $c$), Classifier-Free Guidance (CFG) [15] is widely used to improve alignment by extrapolating from the unconditional

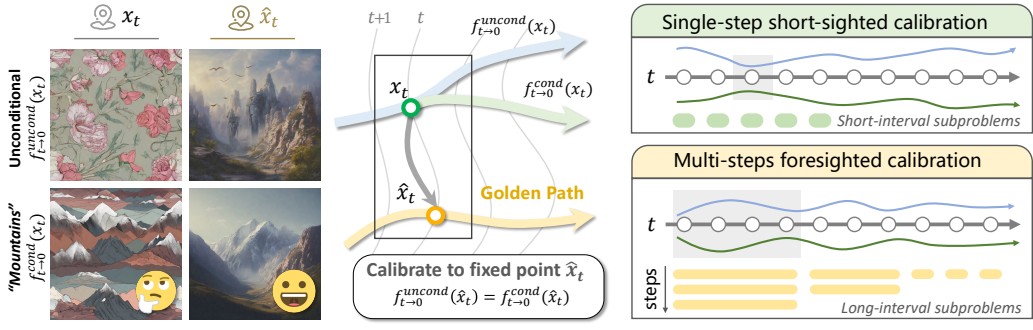

(a) Consistent paths are better    (b) Fixed point framework    (c) **F**ore**s**ight fixed point **g**uidance (FSG)

Figure 1: From left to right: (a) Illustration of the golden path. Latents $\hat{x}_t$ generating mountains unconditionally produce higher-quality images when guided by mountain-related prompts. (b) Unified framework for fixed point iterations. The state $x_t$ is calibrated toward $\hat{x}_t$ on the golden path via fixed point iterations. (c) Proposed foresight guidance (FSG). We enhance efficiency and alignment by conducting fixed point iterations over longer intervals with increased iterations.

noise prediction $\epsilon^u$ towards the conditional prediction $\epsilon^c$, using a guidance scale parameter $w > 1$. The adjusted noise prediction $\epsilon^w$ is computed as:

$$\epsilon^w(x_t) = \epsilon^u(x_t) + w\left(\epsilon^c(x_t) - \epsilon^u(x_t)\right). \tag{4}$$

For brevity, we denote the denoising processes using $\epsilon^u$, $\epsilon^c$, and $\epsilon^w$ as $f^u$, $f^c$, and $f^w$, respectively. CFG++ [5] enhances CFG for manifold preservation through a tunable parameter $\lambda \in [0, 1]$:

$$x_{t-1} = \sqrt{\bar{\alpha}_{t-1}}\phi^\lambda(x_t) + \sqrt{1 - \bar{\alpha}_{t-1}}\epsilon^u(x_t), \quad \phi^\lambda(x_t) = \frac{1}{\sqrt{\bar{\alpha}_t}}(x_t - \sqrt{1 - \bar{\alpha}_t}\epsilon^\lambda(x_t)). \tag{5}$$

Z-sampling [1] and Resampling [20] first adjust $x_t$ to $\hat{x}_t$ through reflection, and then apply an update using $\epsilon^w(\hat{x}_t)$. The reflection in Z-sampling uses $\hat{x}_t = f^w_{t+1\to t} \circ f^u_{t\to t+1}(x_t)$, while Resampling employs $\hat{x}_t = f^w_{t+1\to t} \circ n_{t\to t+1}(x_t)$. Here, $f_{t\to t+1} = \sqrt{\bar{\alpha}_{t+1}}\phi(x_t) + \sqrt{1 - \bar{\alpha}_{t+1}}\epsilon(x_t)$ denotes DDIM inversion [27].

## 3 Methodology

We formalize conditional guidance as calibration toward golden paths, then propose a unified fixed point iteration framework encompassing existing methods (Section 3.1), thus enabling systematic comparison and extension of different design choices. Next, we introduce foresight guidance (FSG), a strategy to address subproblems across longer intervals through multi-step iterations, improving efficacy and efficiency (Section 3.2).

### 3.1 Towards the Golden Path via Fixed Point Iterations

**Consistent denoising paths under conditional and unconditional guidance are golden paths.** In diffusion models that generate images $x_0 \sim p(x \mid c)$ from initial noise $x_T \sim \mathcal{N}(0, \mathbf{I})$, empirical observations suggest that specific denoising paths yield higher-quality images and better alignment under target conditions $c$ [34, 1]. We term these paths as *golden paths*. Our objective is to calibrate the denoising process towards these golden paths.

Let $f^c_{t\to 0}(x_t)$ and $f^u_{t\to 0}(x_t)$ denote the conditional and unconditional denoising processes starting from $x_t$, then the latents $\hat{x}_t$ on golden path satisfy: $f^u_{t\to 0}(\hat{x}_t) = f^c_{t\to 0}(\hat{x}_t)$. For instance, if $\hat{x}_t$ unconditionally generates mountain images, it achieves superior conditional generation performance with mountain-related prompts compared to other latents $x_t$, as visualized in Figure 1(a). The underlying intuition is: when $f^c_{t\to 0}(x_t)$ and $f^u_{t\to 0}(x_t)$ differ significantly, the model requires a *sharp turn*, often sacrificing texture detail and aesthetic quality to achieve conditional alignment. Conversely,

when they align, the model balances both conditional alignment and generation quality. Further discussion is provided in Appendix B.

We therefore aim to calibrate $x_t$ toward $\hat{x}_t$ by solving

$$f_{t\to 0}^u(x_t) = f_{t\to 0}^c(x_t), \quad x_t \in \mathcal{M}_t, \tag{6}$$

where $\mathcal{M}_t$ denotes the probability manifold at timestep $t$.[2]

**A unified framework based on fixed point iteration.** We propose a framework based on fixed point iteration to approximately solve (6), and demonstrate that classifier-free guidance [15] (CFG) and state-of-the-art variants [5, 20, 1] are instances of this unified framework. Specifically, we divide the transition $x_t \to x_{t-1}$ into three phases: (1) Construct a fixed point operator $F(x_t)$ satisfying $F(x_t) \in \mathcal{M}_t$ and $\hat{x}_t = F(\hat{x}_t) \Rightarrow f_{t\to 0}^u(\hat{x}_t) = f_{t\to 0}^u(\hat{x}_t)$. (2) Perform $K$ fixed point iterations from $x_t^{(0)} = x_t$: $x_t^{(k)} = F(x_t^{(k-1)}), k = 1, \cdots, K$ and denote $\hat{x}_t = x_t^{(K)}$. (3) Derive $x_{t-1}$ via the **unconditional** noise prediction $\epsilon^u(\hat{x}_t)$ and a reverse sampler. This workflow can be expressed as:

$$\begin{aligned} \hat{x}_t &= F^{(K)}(x_t) = F \circ \cdots \circ F(x_t), & \triangleright \text{ Calibrate} \\ x_{t-1} &= \text{Sampler}\left(\hat{x}_t, \epsilon^u(\hat{x}_t)\right). & \triangleright \text{ Denoise} \end{aligned} \tag{7}$$

Our framework decouples calibration and denoising, enabling the injection of conditional guidance via the calibration step without binding it to the denoising process. Thus, the key questions of interest for conditional guidance become: (1) What are the design dimensions of fixed point iteration? (2) How can we design along these dimensions for effective guidance? We address the first question below and investigate the second in Section 4.2.

**Design space.** Our fixed point iteration framework identifies four key dimensions: (1) *Consistency interval.* Since the clean semantics $f_{t\to 0}^{u/c}(x_t)$ are inaccessible during denoising, we consider consistency over intervals $a \to b$ as an alternative, i.e., $f_{a\to b}^u(x_t) = f_{a\to b}^c(x_t)$. Longer intervals help to aggregate semantics from more distant timesteps but increase computational cost. (2) *Fixed point operator.* The fixed point operators typically used include the linear operator $x_t = x_t + w(f_{a\to b}^c(x_t) - f_{a\to b}^u(x_t))$ and the backward-forward operator $x_t = f_{a\to b}^{u(-1)} \circ f_{a\to b}^c(x_t)$. (3) *Guidance strength/scheduler.* We can schedule the magnitude of updates for different timesteps. Similar to the learning rate, inappropriate settings can lead to slow convergence or deviation from the data manifold. (4) *Number of iterations $K$.* Ideally, the fixed point algorithm converges linearly w.r.t. $K$, i.e., $\|\hat{x}_t - x_t\| = \mathcal{O}(\rho^K)$, where $\rho \in (0, 1)$ is the spectral radius of $F'$.

Surprisingly, we find that CFG and its variants can be incorporated into our fixed point framework despite originating from different derivations, which provides us with a unified perspective. We present their design choices in Table 1.

*Sketch of the unification. (detailed in Appendix A)* We isolate the unconditional noise in the update of DDIM [27] and unify CFG [15] and CFG++ [5] as linear fixed point operators in the interval $[t-1, t]$, using the equivalence of $\epsilon^c(x_t) = \epsilon^u(x_t)$ to $f_{t\to t-1}^c(x_t) = f_{t\to t-1}^u(x_t)$. Approximating $f_{t+1\to t}^{u(-1)}$ by $f_{t\to t+1}^u$ or $n_{t\to t+1}$, we equate reflective sampling in Z-sampling [1] and Resampling [20] to backward-forward operators. Further nested CFG updates yield the results shown in Table 1.

**Comparison of design choices.** (1) *Iteration strength scheduler for CFG ($w\xi_t$) and CFG++ ($\lambda\tilde{\xi}_t$):* Compared to CFG, CFG++ provides more stable iteration strength during the critical early generation stage, avoiding drastic decay. This explains the improved stability of CFG++ over CFG. (2) *Fixed point operators:* Compared to the linear operators $x_t + w\xi_t\Delta\epsilon(x_t)$ used in CFG and CFG++, we observe that the forward-backward operator $f_{a\to b}^{u(-1)} \circ f_{a\to b}^c(x_t)$ with larger consistency intervals exhibits lower empirical contraction rates. This facilitates faster convergence, albeit at the cost of increased model evaluations. A more detailed discussion is provided in Appendix A.5.

**Extension of design choices.** The unified fixed point framework allows us to directly upgrade existing algorithms by increasing iteration counts. We extend CFG and CFG++ to CFG / CFG++ $\times K$, where $K$ denotes the number of iterations (Algorithm 2,3). However, this approach leads to a substantial increase in model evaluations. Therefore, we propose *Foresight Guidance*, which balances efficiency and effectiveness by prioritizing longer intervals and more iterations in the early stages.

---

[2]This constraint aims to prevent inaccurate score estimation off the manifold. Since satisfying this constraint strictly is not the primary focus of this paper, we simply follow empirical settings, e.g., use smaller step sizes.

Table 1: Unification of different guidance methods as fixed point iterations via $x^{k+1} = F(x^k)$, aiming for $f^u_{a \to b}(x_t) = f^c_{a \to b}(x_t)$. Multi-iter. indicates whether the original algorithm supports multi-step iterations. Abbreviations: (1) $w > 1, \xi_t = \sqrt{1 - \bar{\alpha}_t} - \sqrt{\alpha_t - \bar{\alpha}_t}$ for CFG; (2) $\lambda \in [0,1], \tilde{\xi}_t = \sqrt{1 - \bar{\alpha}_t}$ for CFG++; (3) $f^\gamma$ denotes denoising with noise $\epsilon^u + \gamma(\epsilon^c - \epsilon^u)$, $\Delta\epsilon(\cdot)$ denotes $\epsilon^c(\cdot) - \epsilon^u(\cdot)$, and id denotes the identity mapping.

| Methods | Fixed point operator $F(x_t)$ | Interval $(a \to b)$ | Multi-iter. |
|---|---|---|---|
| CFG [15] | $x_t - w\xi_t\Delta\epsilon(x_t)$ | $t \to t-1$ | ✗ |
| CFG++ [5] | $x_t - \lambda\tilde{\xi}_t\Delta\epsilon(x_t)$ | $t \to t-1$ | ✗ |
| Z-sampling [1] | $(\text{id} - w\xi_t\Delta\epsilon) \circ f^\gamma_{t+1 \to t} \circ f^u_{t \to t+1}(x_t)$ | $t+1 \to t-1$ | ✗ |
| Resampling [20] | $(\text{id} - w\xi_t\Delta\epsilon) \circ f^\gamma_{t+1 \to t} \circ n_{t \to t+1}(x_t)$ | $t+1 \to t-1$ | ✓ |
| FSG (ours) | $(\text{id} - \lambda\tilde{\xi}_t\Delta\epsilon) \circ f^u_{t-\Delta t \to t} \circ f^\gamma_{t \to t-\Delta t}(x_t)$ | $t \to t - \Delta t$ | ✓ |

## 3.2 Foresight Guidance

We aim to minimize the gap between conditional and unconditional paths. Existing methods typically divide the problem into $T$ subproblems with intervals of $t/t+1 \to t-1$, each solved with one step of fixed point iteration (illustrated in Figure 1 (b)). While each subproblem is relatively simple, the large number of subproblems limits efficiency, especially for high-overhead fixed point operators such as backward-forward operators. Moreover, small intervals limit the benefit of calibration, as only the semantics of neighboring timesteps can be obtained for guidance.

To improve efficiency, instead of allocating one iteration for each small interval, we can allocate multiple iterations for fewer long intervals. We theoretically demonstrate that the single-step short-interval strategy is typically suboptimal. Given the total iteration budget $N$ and timesteps $T$, we uniformly divide $f^u_{T \to 0} = f^c_{T \to 0}$ into $M$ subproblems ($f^u_{iT/M \to (i-1)T/M} = f^c_{iT/M \to (i-1)T/M}, i \in [M]$), solving each only at timesteps $iT/M$ with $N/M$ fixed point iterations (assuming $N/M, T/M \in \mathbb{Z}$). The $M = T$ case represents the short-interval single-step strategy.

---

**Algorithm 1:** Foresight Guidance (FSG)

**Input** : Initial noise $x_T$, Condition $c$, Timesteps $T$, Iteration set $\mathcal{S} = \{(t_i, K_i, \Delta t_i)\}_{i=1}^M$, Strengths $\gamma > 1, \lambda \in [0,1]$.

**Output** : Generated image $x_0$

1 **for** $t \leftarrow T$ **to** 1 **do**
2    **if** $(t, K, \Delta t) \in \mathcal{S}$ **then**
3      *Foresight Fixed Point Calibration*;
4      **for** $k \leftarrow 1$ **to** $K$ **do**
5        $x_{t-\Delta t} = f^\gamma_{t \to t-\Delta t}(x_t)$;
6        $x_t = f^u_{t-\Delta t \to t}(x_{t-\Delta t})$;
7      **end**
8    **end**
9    *CFG++ Calibration*;
10    $\hat{x}_t = x_t - \lambda\tilde{\xi}_t(\epsilon^c(x_t) - \epsilon^u(x_t))$;
11    *Denoising Step*;
12    $x_{t-1} = \text{Sampler}(\hat{x}_t, \epsilon^u(x_t))$;
13 **end**
14 **return** $x_0$

---

**Theorem 1.** *(Detailed in Appendix C) Let $\mathcal{L} = \frac{1}{T}\sum_{t=1}^T \|\epsilon^c(\hat{x}_t) - \epsilon^u(\hat{x}_t)\|_2^2$ denote the average gap over calibrated trajectories $\hat{x}_t \in \mathbb{R}^d$, with $B$ as the Euclidean norm bound for $\hat{x}_t$ and $\epsilon^{c/u}(\hat{x}_t)$, $L$ as the smoothness constant of $\epsilon^{c/u}(\cdot)$, and $r \in (0,1)$ as the upper bound of the contraction rate of $F_i$, $i \in [M]$. Under mild assumptions (Appendix C), there exists a constant $C > 0$ such that*

$$\mathcal{L} = \frac{1}{T}\sum_{t=1}^T \|\epsilon^c(\hat{x}_t) - \epsilon^u(\hat{x}_t)\|_2^2 \leq B^2 \left( Cr^{\frac{2N}{M}} + \frac{2L^2}{M^2} \right). \tag{8}$$

Setting the derivative of the right-hand side to zero yields the optimal $M^*$ that minimizes the upper bound. The optimal $M^*$ is typically not $T$, indicating that performing fixed point iterations at every timestep is unnecessary. Key insights include: (1) Smaller $L$ (smoother noise predictors) reduces $M^*$, favoring fewer, longer-interval subproblems; (2) Sufficient computational resources ($N \to \infty$) drive $M \to T$, recovering the short-interval subproblems. In practice, limited resources suggest using moderate interval sizes to enhance fixed point solving efficiency. Intuitively, longer intervals provide stronger guidance during early denoising stages, where short-term approximations ($f^{u/c}_{t \to t-1}$) yield limited benefits due to insufficient estimates of the clean image. Thus, generating prototypes through

Table 2: The quantitative results on the SDXL model with NFE = 50, 100, 150 (Time denotes seconds per image, ↑ denotes higher is better, The best results under the same NFE are bolded).

| Datasets | | | DrawBench [25] | | | | Pick-a-Pic [17] | | | |
|---|---|---|---|---|---|---|---|---|---|---|
| Method | NFE | Time | IR↑ | HPSv2↑ | AES↑ | CLIP↑ | IR↑ | HPSv2↑ | AES↑ | CLIP↑ |
| CFG | 50 | 6.71 | 59.02 | 28.73 | 6.07 | 32.29 | 82.14 | 28.46 | **6.73** | 33.53 |
| CFG++ | 50 | 6.82 | 65.21 | 28.98 | **6.08** | 32.60 | 89.75 | 28.72 | 6.67 | 33.86 |
| Z-Sampling | 50 | 6.66 | 72.75 | 29.08 | 6.00 | 32.59 | 96.77 | 28.68 | 6.59 | 33.97 |
| Resampling | 50 | 6.48 | 59.99 | 28.80 | 5.99 | 32.21 | 82.65 | 28.46 | 6.61 | 33.46 |
| FSG (ours) | 50 | 6.77 | **82.81** | **29.42** | 6.01 | **32.65** | **98.59** | **28.89** | 6.60 | **34.32** |
| CFG×2 | 100 | 12.51 | 77.71 | 29.36 | **6.06** | 32.44 | 96.06 | 28.84 | 6.64 | 34.13 |
| CFG++×2 | 100 | 12.60 | 79.42 | 29.42 | 6.01 | 32.61 | 99.90 | 29.00 | 6.61 | 34.18 |
| Z-Sampling | 100 | 12.61 | 77.46 | 29.26 | 6.03 | 32.39 | 94.98 | 28.79 | 6.61 | 34.01 |
| Resampling | 100 | 12.49 | 77.26 | 29.12 | 6.00 | 32.46 | 79.36 | 28.61 | 6.02 | 33.61 |
| FSG (ours) | 100 | 12.56 | **84.12** | **29.54** | 6.02 | **32.76** | **102.82** | **29.05** | **6.66** | **34.30** |
| CFG×3 | 150 | 18.47 | 83.56 | **29.51** | 5.95 | 32.66 | 102.13 | 29.04 | 6.61 | **34.28** |
| CFG++×3 | 150 | 18.47 | 82.58 | 29.45 | 5.93 | 32.66 | 103.32 | **29.05** | 6.57 | 34.20 |
| Z-Sampling | 150 | 18.26 | 78.35 | 29.40 | **6.06** | 32.43 | 97.25 | 28.90 | **6.67** | 34.20 |
| Resampling | 150 | 18.26 | 79.98 | 29.23 | 6.05 | 32.32 | 87.48 | 28.70 | 6.59 | 33.49 |
| FSG (ours) | 150 | 18.49 | **88.18** | 29.44 | 5.96 | **32.70** | **104.86** | 29.04 | 6.65 | **34.28** |

extended conditional denoising processes $f^c_{t\to t-\Delta t}$ and preserving information into $\hat{x}_t$ via the inverse ODE $f^u_{t-\Delta t\to t}$ appears more effective.

**Practical design of foresight guidance.** We integrate these design choices into Foresight Guidance (FSG), as outlined in Algorithm 1. We perform multi-step fixed-point iterations with long time intervals at specific timesteps, parameterized by a set $\mathcal{S} = (t_i, K_i, \Delta t_i)^M_{i=1}$, where each tuple denotes the starting timestep, number of iterations, and interval length, respectively. To reduce computational overhead, we employ a forward-backward operator with a single-step ODE solver (DDIM) for both $f^\gamma_{t\to t-\Delta t}$ and $f^u_{t-\Delta t\to t}$. As most image semantics are determined in the early stages of diffusion [31], we allocate more iterations ($K_i$) and longer intervals ($\Delta t_i$) during these phases, following a ratio of approximately 3:2:1 across early, middle, and late stages. Outside these scheduled iterations, we apply CFG++ to maintain stable guidance strength and avoid oscillations. This allows early generation stages to benefit from the semantics of future states, motivating the term *foresight guidance*.

# 4 Experiments

## 4.1 Experimental Setup

**Datasets.** We assess generation performance across four benchmark datasets: DrawBench [25], Pick-a-Pic [17], Geneval [10], and PartiPrompts [32]. Detailed experimental setups are provided in Appendix D.1, and results for PartiPrompts are included in Appendix D.2.

**Metrics.** To evaluate results, we employ IR [30] and HPSv2 [29] as human preference metrics, ClipScore [12] for prompt alignment assessment, and AES [26] for aesthetic quality analysis.

**Baselines.** Under our unified fixed point iteration framework, we systematically analyze five methods: CFG [15], CFG++ [5], Z-sampling [1], Resampling [20], and our proposed FSG. Leveraging the fixed point perspective, we enhance CFG and CFG++ by increasing iterations (denoted as ×2/×3), a novel extension beyond prior literature [15, 5]. Experiments primarily adopt configurations from original studies. The fixed point interval in FSG is set within $[0.02T, 0.125T]$, with larger intervals and iterations allocated to early steps. Implementation details are provided in Appendix D.1.

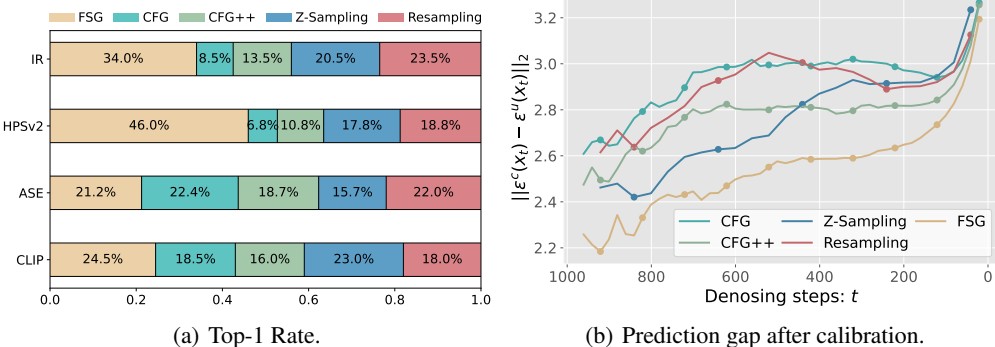

| (a) Top-1 Rate. | (b) Prediction gap after calibration. |

Figure 2: Left: Proportion of samples that outperform the other four methods (Top-1 rate). Right: Prediction gap during denoising, indicating the effectiveness of the fixed point iteration. Dataset: DrawBench, NFE: 50, see Appendix D.3 and D.4 for more results.

Table 3: Quantitative results on Geneval dataset. Model: SDXL; NFE: 50 (CFG), 150 (others).

| Method | Overall↑ | Single Object↑ | Two Object↑ | Counting↑ | Colors↑ | Position↑ | Color Attribution↑ |
|---|---|---|---|---|---|---|---|
| CFG | 48.39 % | 97.50 % | 61.62 % | 22.50 % | 78.72 % | **14.00 %** | 16.00 % |
| CFG×3 | 55.94 % | 98.75 % | 75.76 % | 40.00 % | 85.11 % | 8.00 % | **28.00 %** |
| CFG++×3 | 56.03 % | 97.50 % | 78.79 % | 45.00 % | 81.91 % | 10.00 % | 23.00 % |
| Z-sampling | 56.70 % | **100.00 %** | 75.76 % | **46.25 %** | **86.17 %** | 12.00 % | 20.00 % |
| Resampling | 56.65 % | **100.00 %** | **84.85 %** | 40.00 % | 84.04 % | 7.00 % | 24.00 % |
| FSG (ours) | **57.95 %** | **100.00 %** | 79.80 % | 43.75 % | **86.17 %** | 12.00 % | **28.00 %** |

## 4.2 Experimental Results

**Quantitative analysis.** We evaluate the performance of different methods at NFE (number of function evaluations) of 50, 100, and 150 using SDXL with the DDIM sampler, as shown in Table 2. The following discussion analyzes design choices within the fixed point iteration framework:

1. **Extended intervals and increased iterations enhance efficiency and alignment of FSG.** FSG demonstrates superior performance across datasets and NFEs, with particularly notable improvements at lower NFEs (50, 100). As shown in Table 2 and the Top-1 rate in Figure 2, FSG improves IR by 10.06 and achieves a Top-1 rate of 34%, while improving HPSv2 by 0.34 with a Top-1 rate of 46%. This improvement is attributed to more effective fixed point solving. The reduced fixed point error in Figure 2 confirms that balanced subproblem decomposition enables better convergence and generation quality, consistent with Theorem 1. Additionally, longer intervals enhance semantic guidance and prompt-alignment in FSG, as evidenced by consistent CLIPScore and IR improvements across all NFEs.

2. **Existing methods benefit from increased iterations.** Within the fixed point framework, CFG and CFG++ outperform vanilla CFG when the number of iterations is increased (denoted as ×2/×3). This demonstrates the potential of our framework to enable test-time scaling for improved performance through additional inference resources, preferable to simply increasing inference steps (see Appendix D.5).

3. **Strength schedulers and fixed point operators.** CFG++ achieves marginal gains over CFG through more stable intensity scheduling. Backward-forward operators with small intervals show limited improvement compared to linear operators at higher NFEs due to insufficient guidance, necessitating longer intervals for adequate semantic guidance.

**Qualitative analysis.** We present qualitative results in Figure 3, comparing the performance of FSG with baseline methods. FSG achieves both high image aesthetics and strong adherence to prompt requirements. By leveraging longer intervals, FSG achieves more precise guidance for generating fine

Table 4: Quantitative results on different models and samplers. Dataset:Pick-a-pic; NFE: 50 (CFG), 150 (others); see Appendix D.6 for more results.

| Models | SD-2.1 [24], DDIM | | | | Hunyuan-DiT [18], DDIM | | | | SDXL, DDPM [14] | | | |
| Method | IR↑ | HPSv2↑ | AES↑ | CLIP↑ | IR↑ | HPSv2↑ | AES↑ | CLIP↑ | IR↑ | HPSv2↑ | AES↑ | CLIP↑ |
|---|---|---|---|---|---|---|---|---|---|---|---|---|
| CFG | -62.83 | 25.51 | 5.88 | 30.38 | 115.63 | 29.00 | 6.82 | 33.09 | 73.57 | 28.42 | 6.73 | 33.62 |
| CFG ×3 | 1.08 | 27.25 | 5.92 | 32.50 | 115.32 | 28.98 | 6.50 | 32.88 | 91.37 | **28.69** | 6.62 | 33.73 |
| CFG++×3 | 3.98 | 27.24 | 5.96 | 32.51 | 115.63 | 29.03 | 6.63 | 32.97 | 89.17 | 28.79 | 6.60 | 33.60 |
| Z-sampling | 3.65 | 27.24 | 6.07 | 32.60 | 128.72 | 29.23 | 6.72 | 33.42 | 90.35 | 28.63 | **6.65** | 33.58 |
| Resampling | 8.07 | 27.03 | 5.85 | 32.31 | 117.65 | 29.28 | **6.73** | 33.18 | 90.92 | 28.64 | **6.65** | 33.73 |
| FSG (ours) | **16.26** | **27.60** | **6.10** | **32.80** | **132.88** | **29.37** | 6.68 | **33.48** | **91.53** | 28.56 | **6.65** | **33.79** |

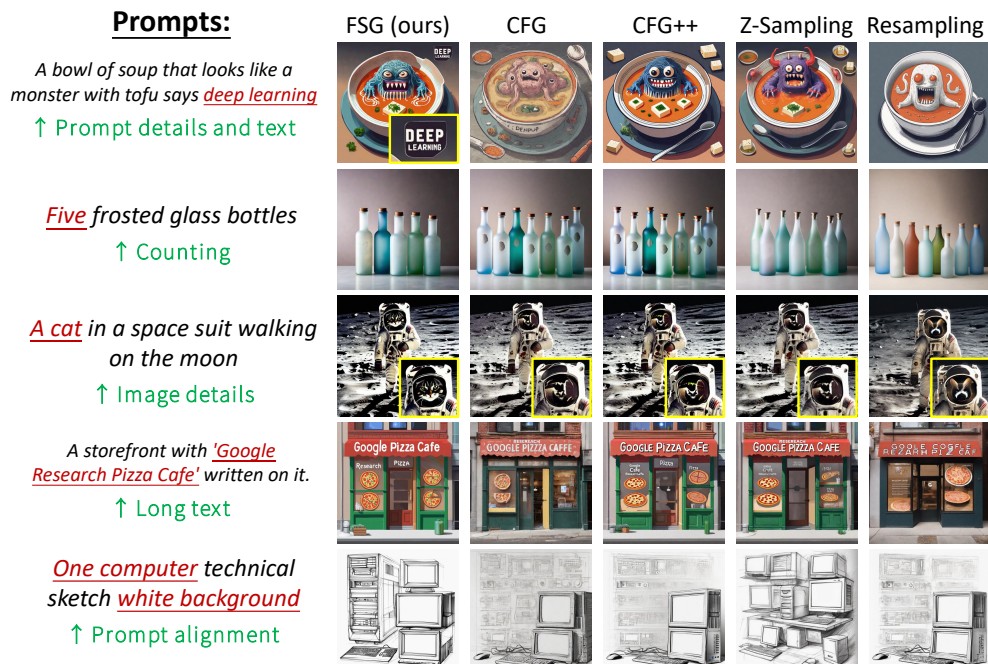

Figure 3: Results of qualitative analysis. Our proposed FSG demonstrates effectiveness in several dimensions including text, details, and counting. We present more examples in Appendix E.

details (e.g., cat faces, text). Furthermore, FSG-generated images exhibit fewer structural or visual artifacts and align more closely with human preferences. These results highlight the robustness and versatility of FSG in addressing complex image generation tasks. We present additional visualizations and discuss failure cases in Appendix E.

**Object-focused evaluation.** We assess fine-grained instruction compliance on Geneval [10], as shown in Table 3. FSG significantly addresses the deficiencies of vanilla CFG in counting accuracy (+23.75%) and two-object generation (+23.23%), achieving state-of-the-art overall performance. Notably, CFG/CFG++ with increased fixed point iterations also demonstrate enhanced alignment, underscoring the benefits of fixed point iterations.

**Class conditional generation.** To investigate whether enhanced fixed point iterations reduce diversity, we conduct experiments on the ImageNet [6] $256 \times 256$ conditional generation task using DiT [21] models, generating 1K images per class (totaling 50K images). We report FID [13] and Vendi scores [7] at NFE=25/50. As shown in Table 5, FSG and CFG/CFG++×2 at NFE=50 both improve generation quality and diversity, indicating that fixed point iterations do not cause mode collapse. We hypothesize this occurs because the iterations are performed locally in the neighborhood of $x_t$, thereby improving quality while preserving randomness.

Table 5: Quantitative results on ImageNet $256 \times 256$. Model: DiT.

| NFE | 25 | | 50 | |
|---|---|---|---|---|
| Methods | FID ↓ | Vendi ↑ | FID ↓ | Vendi ↑ |
| CFG (×2) | 17.81 | 3.44 | 14.69 | 3.79 |
| CFG++ (×2) | 13.27 | 3.91 | 8.85 | 4.43 |
| Resample | 17.54 | 3.50 | 9.05 | 4.47 |
| Z-sampling | 19.89 | 3.40 | 8.62 | 4.64 |
| FSG (ours) | **10.56** | **4.73** | **7.91** | **5.79** |

Table 6: Synergistic effects of FSG and noise optimization model (NPNet) Dataset: Pick-a-Pic.

| Methods | IR↑ | HPSv2↑ | AES↑ | CLIP↑ |
|---|---|---|---|---|
| CFG50 | 82.14 | 28.46 | **6.73** | 33.53 |
| FSG50 | 98.59 | 28.89 | 6.60 | **34.32** |
| FSG100 | 102.82 | 29.05 | 6.66 | 34.30 |
| NPNet | 91.66 | 28.60 | 6.70 | 33.57 |
| +FSG50 | **112.64** | 29.04 | 6.54 | 34.09 |
| +FSG100 | 111.83 | **29.15** | 6.57 | 34.13 |

Table 7: Synergistic effects of FSG and preference alignment model (SPO). Dataset: Pick-a-pic.

| Method | IR↑ | HPSv2↑ | AES↑ | CLIP↑ |
|---|---|---|---|---|
| CFG50 | 82.14 | 28.46 | 6.73 | 33.53 |
| FSG100 | 102.82 | 29.05 | 6.66 | 34.30 |
| SPO | 111.86 | 29.08 | 6.91 | 33.22 |
| +FSG50 | 115.86 | 29.16 | 6.91 | 33.12 |
| +FSG100 | **117.93** | **29.20** | **6.93** | 33.24 |
| +FSG150 | 116.49 | 28.74 | 6.85 | **33.30** |

**Prompt:**
*3D Pac Man in real life*

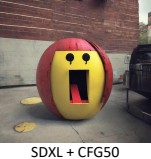
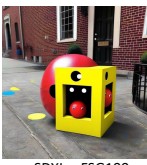

SDXL + CFG50

SDXL + FSG100

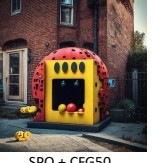
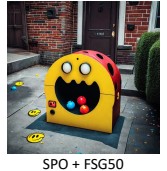
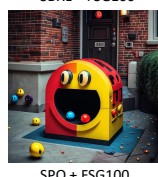

SPO + CFG50

SPO + FSG50

SPO + FSG100

Figure 4: FSG provides better guidance on SPO and progressively improves generated images.

**Different models and samplers.** Results across models (SD2.1 [24] and Hunyuan-DiT [18]) and samplers (DDPM [14]) are presented in Table 4. Compared to other methods, FSG demonstrates greater improvement on the weaker SD2.1 baseline (IR: +8.19; HPSv2: +0.35) while still providing additional quality gains for the state-of-the-art Hunyuan-DiT model (IR: +4.16; HPSv2: +0.09). As a sample-agnostic framework, our method naturally extends to stochastic samplers without additional derivations, where fixed point iteration similarly improves prompt alignment.

**Synergy with orthogonal methods.** As fixed point algorithms are sensitive to initial values, we explore the synergistic effects between FSG and orthogonal approaches, including preference-aligned models [19] and noise optimization methods [34]. SPO [19] improves the noise predictions of the diffusion model via preference fine-tuning, while NPNet [34] employs lightweight networks to tailor initial noise to a given prompt. Both strategies supply better initial values to the fixed-point iteration, facilitating coarse-to-fine calibration. Experiments conducted with NPNet [34] (Table 6) and SPO [19] (Table 7) demonstrate that FSG alone outperforms noise optimization , but falls short of preference-aligned models. When FSG is integrated with either approach, synergistic improvements are observed, leading to further gains in aesthetic quality and prompt alignment (IR: 112.64 with NPNet; 117.93 with SPO). Figure 4 illustrates the progressive quality enhancement when combining FSG with SPO, confirming the compatibility of our framework with orthogonal approaches.

**Ablation studies.** Table 8 summarizes the impactful design choices at NFE = 50/150. Key findings: (1) *Consistency intervals* (×2/½): At low NFEs, appropriate interval selection provides more effective guidance than short-interval strategies. Overly large intervals introduce distant guidance that exceeds the current optimization stage, hindering fixed point iteration convergence. (2) *Guidance strength*: As shown in Figure 5, FSG adapts to $\lambda \in [0.4, 1.3]$ (beyond the $[0.5, 1]$ range in CFG++). Fixed point iteration narrows the conditional/unconditional gap while suppressing quality degradation from excessive guidance. The scheduler of CFG++ prevents premature intensity decay, making it suitable for our approach. (3) *Iterations* (×2/½): While 2-3 iterations suffice for high-quality results, excessive iterations introduce computational overhead and risk divergence from the data manifold. (4) *Timestep prioritization*: Early-stage (($\frac{2}{3}T$,$T$]) foresighted guidance drives quality gains, while later timestep calibrations refine details.

Table 8: Ablation study on different choices (metrics differences shown). Dataset: Pick-a-Pic.

| Method | IR↑ | HPSv2↑ | AES↑ | CLIP↑ |
|---|---|---|---|---|
| FSG50 | 98.59 | 28.89 | 6.60 | 34.32 |
| Interval×½ | -8.20 | -0.04 | 0.01 | -0.13 |
| Interval×2 | -2.40 | -0.12 | -0.01 | -0.23 |
| FSG150 | 104.86 | 29.04 | 6.65 | 34.28 |
| Iterations×½ | -6.16 | -0.21 | +0.03 | -0.23 |
| Iterations×2 | -2.41 | -0.50 | -0.13 | -0.07 |
| Early | -4.82 | -0.19 | -0.08 | -0.14 |
| Early+Mid | -2.33 | -0.12 | -0.07 | -0.14 |
| w.o. CFG++ | -4.78 | -0.08 | +0.03 | -0.03 |

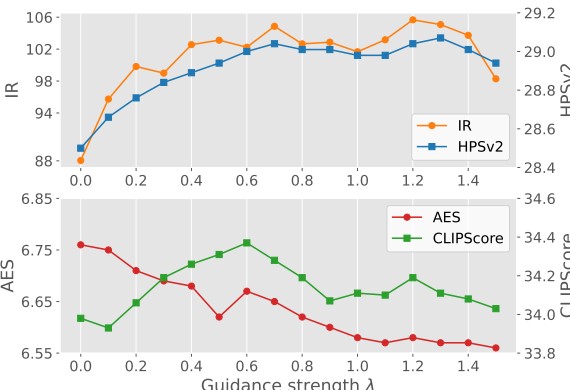

Figure 5: Guidance strength analysis. FSG shows robustness on $\lambda \in [0.4, 1.3]$. Dataset: Pick-a-Pic; NFE:150.

## 5 Related Works

**Classifier-Free Guidance [15] (CFG)** is crucial for modern text-to-image diffusion models, yet its mechanisms remain unclear. While initially framed as sampling from $\tilde{p}(x|c) \propto p(x)p(c|x)^w$, this interpretation has proven inconsistent with implementation behaviors [4]. Recent works pursue distinct objectives: sampling from $\tilde{p}(x|c) \propto p(x|c)R(x,c)$ [9], minimizing score distillation sampling loss [5], functioning as a predictor–corrector [2], and satisfying Fokker-Planck equations [33]. However, these approaches' entanglement with diffusion sampling dynamics impedes both theoretical progress and unified method comparison. Our fixed point iteration framework circumvents this coupling through a sampling-independent interpretation.

**The golden path phenomenon** states that specific noise yields superior generation outcomes [23, 34]. Empirical observations demonstrate that noise capable of unconditionally generating an image matching condition $c$ achieves enhanced performance when guided by $c$ [23, 1]. This observation has motivated various approaches, including optimizing initial noise [11, 23], neural network-based noise prediction [34], and progressive refinement during inference [1]. While effective, these techniques incur significant computational overhead. To the best of our knowledge, our work presents the first established connection between this phenomenon and CFG.

## 6 Conclusion

In this work, we introduce a unified framework for classifier-free guidance (CFG) in text-to-image diffusion models by reinterpreting conditional generation as a calibration process toward a golden path. First, we propose fixed point iteration as a methodological tool to enforce latent consistency between conditional and unconditional outputs, establishing a sampling-independent framework with broader design applicability. We unify existing CFG and its variants as short-interval single-step approaches under this framework, which are theoretically inefficient. Second, we present Foresight Guidance (FSG), a multi-step iteration paradigm that reduces subproblems while enabling long-horizon guidance, achieving an enhanced alignment-quality balance. Comprehensive experiments demonstrate the superiority of FSG over state-of-the-art methods. We anticipate that our unified perspective will catalyze advancements in adaptive, efficient diffusion guidance strategies.

## Acknowledgements

We would like to thank the anonymous reviewers for their valuable suggestions on theory and experiments. This work is supported by the Project of Hetao Shenzhen-HKUST Innovation Cooperation Zone HZQBKCZYB-2020083.

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

# A   Unified Framework

In this section, we establish a theoretical framework that unifies guidance methods under the perspective of fixed-point iterations. Through systematic examination of Classifier-Free Guidance (CFG), its variant CFG++, and reflective sampling techniques including Z-Sampling and Resampling, we demonstrate how these approaches achieve conditional generation through distinct yet analogous fixed point operators. Our framework decouples guidance to denoising-agnostic calibration steps, facilitating the design of guidance algorithms.

## A.1   Classifier-Free Guidance (CFG) [15]

Classifier-Free Guidance (CFG) enhances conditional generation by interpolating conditional and unconditional noise predictions during the diffusion process. Our analysis reveals that CFG's forward process corresponds to a linear fixed-point operator with consistency intervals spanning $t \to t - 1$, requiring 1 NFE per iteration.

To formalize this, we begin with the DDIM update step governed by the predicted noise $\epsilon(x_t)$:

$$x_{t-1} = \frac{\sqrt{\bar{\alpha}_{t-1}}}{\sqrt{\bar{\alpha}_t}} \left[ x_t - \sqrt{1 - \bar{\alpha}_t}\epsilon(x_t) \right] + \sqrt{1 - \bar{\alpha}_{t-1}}\epsilon(x_t), \tag{9}$$

where $\bar{\alpha}_t = \prod_{i=1}^t \alpha_i = \alpha_t \bar{\alpha}_{t-1}$. Through algebraic rearrangement, we derive an equivalent representation that better elucidates the noise dependency:

$$x_{t-1} = \frac{\sqrt{\bar{\alpha}_{t-1}}}{\sqrt{\bar{\alpha}_t}} x_t + \left( \sqrt{1 - \bar{\alpha}_{t-1}} - \frac{\sqrt{\bar{\alpha}_{t-1}}}{\sqrt{\bar{\alpha}_t}} \sqrt{1 - \bar{\alpha}_t} \right) \epsilon(x_t)$$

$$= \frac{\sqrt{\bar{\alpha}_{t-1}}}{\sqrt{\bar{\alpha}_t}} x_t + \left( \sqrt{1 - \bar{\alpha}_{t-1}} - \frac{\sqrt{1 - \bar{\alpha}_t}}{\sqrt{\alpha_t}} \right) \epsilon(x_t). \tag{10}$$

CFG introduces a rescaled noise prediction to amplify conditional guidance:

$$\epsilon^w(x_t) = \epsilon^u(x_t) + w(\epsilon^c(x_t) - \epsilon^u(x_t)), \tag{11}$$

where $w > 1$ modulates guidance strength. Substituting $\epsilon^w(x_t)$ into the update rule yields:

$$x_{t-1} = \frac{\sqrt{\bar{\alpha}_{t-1}}}{\sqrt{\bar{\alpha}_t}} x_t + \left( \sqrt{1 - \bar{\alpha}_{t-1}} - \frac{\sqrt{1 - \bar{\alpha}_t}}{\sqrt{\alpha_t}} \right)$$

$$\times \left[ \epsilon^u(x_t) + w(\epsilon^c(x_t) - \epsilon^u(x_t)) \right]. \tag{12}$$

This formulation admits an insightful reinterpretation as a two-step process. First, the latent variable $\hat{x}_t$ is calibrated via:

$$\hat{x}_t = x_t - w \left( \sqrt{1 - \bar{\alpha}_t} - \sqrt{\alpha_t - \bar{\alpha}_t} \right) (\epsilon^c(x_t) - \epsilon^u(x_t)), \tag{13}$$

followed by an unconditional denoise:

$$x_{t-1} = \frac{\sqrt{\bar{\alpha}_{t-1}}}{\sqrt{\bar{\alpha}_t}} \hat{x}_t + \left( \sqrt{1 - \bar{\alpha}_{t-1}} - \frac{\sqrt{1 - \bar{\alpha}_t}}{\sqrt{\alpha_t}} \right) \epsilon^u(x_t). \tag{14}$$

The iterative variant CFG$\times K$ is detailed in Algorithm 2.

## A.2   CFG++ [5]

CFG++ derives conditional guidance through manifold constraints formulated from an inverse problem perspective, closely aligned with CFG in implementation. Our analysis demonstrates that CFG++ similarly corresponds to a linear fixed-point operator iteration but diverges from CFG solely in its strength scheduling strategy, retaining the consistency interval $t \to t - 1$ at a computational cost of 1 NFE per iteration.

CFG++ adjust the guidance strength through parameter $\lambda$ applied to the noise difference term:

$$x_{t-1} = \frac{\sqrt{\bar{\alpha}_{t-1}}}{\sqrt{\bar{\alpha}_t}} \left\{ x_t - \sqrt{1 - \bar{\alpha}_t}[\epsilon^u(x_t) + \lambda(\epsilon^c(x_t) - \epsilon^u(x_t))] \right\}$$

$$+ \sqrt{1 - \bar{\alpha}_{t-1}}\epsilon^u(x_t). \tag{15}$$

This admits similar calibration-interpretation, where the latent calibration becomes:

$$\hat{x}_t = x_t - \lambda\sqrt{1 - \bar{\alpha}_t}(\epsilon^c(x_t) - \epsilon^u(x_t)), \tag{16}$$

followed by the same unconditional update:

$$x_{t-1} = \frac{\sqrt{\bar{\alpha}_{t-1}}}{\sqrt{\bar{\alpha}_t}}\hat{x}_t + \left(\sqrt{1 - \bar{\alpha}_{t-1}} - \frac{\sqrt{1 - \bar{\alpha}_t}}{\sqrt{\bar{\alpha}_t}}\right)\epsilon^u(x_t). \tag{17}$$

Both CFG and CFG++ exhibit fixed-point behavior through linear operators:

$$F_{\text{CFG}}(x_t) = x_t + w\xi_t(\epsilon^c(x_t) - \epsilon^u(x_t)), \tag{18}$$

$$F_{\text{CFG++}}(x_t) = x_t + \lambda\tilde{\xi}_t(\epsilon^c(x_t) - \epsilon^u(x_t)), \tag{19}$$

where $\xi_t = \left(\sqrt{1 - \bar{\alpha}_t} - \sqrt{\alpha_t - \bar{\alpha}_t}\right), \tilde{\xi}_t = \sqrt{1 - \bar{\alpha}_t}$ respectively. Note that fixed-point $F(x_t) = x_t$ implies consistency $\epsilon^c(x_t) = \epsilon^u(x_t)$, enforcing alignment between conditional and unconditional update paths: $f_{t\to t-1}^u(x_t) = f_{t\to t-1}^c(x_t)$. The iterative variant CFG++$\times K$ is detailed in Algorithm 3.

| **Algorithm 2:** CFG$\times K$ | **Algorithm 3:** CFG++$\times K$ |
|---|---|
| **Input** : Initial noise $x_T$, Condition $c$, Timesteps $T$, Iterations $K$, Strength $w > 1$. | **Input** : Initial noise $x_T$, Condition $c$, Timesteps $T$, Iterations $K$, Strength $\lambda \in [0, 1]$. |
| **Output :** Generated image $x_0$ | **Output :** Generated image $x_0$ |
| 1 **for** $t \leftarrow T$ **to** 1 **do** | 1 **for** $t \leftarrow T$ **to** 1 **do** |
| 2    $x_t^{(0)} = x_t$; | 2    $x_t^{(0)} = x_t$; |
| 3    *CFG Fixed Point Calibration*; | 3    *CFG++ Fixed Point Calibration*; |
| 4    **for** $k \leftarrow 1$ **to** $K$ **do** | 4    **for** $k \leftarrow 1$ **to** $K$ **do** |
| 5      $\xi_t = \sqrt{1 - \bar{\alpha}_t} - \sqrt{\alpha_t - \bar{\alpha}_t}$; | 5      $\tilde{\xi}_t = \sqrt{1 - \bar{\alpha}_t}$; |
| 6      $x_t^{(k)} = x_t^{(k-1)} - w\xi_t\Delta\epsilon(x_t^{(k-1)})$; | 6      $x_t^{(k)} = x_t^{(k-1)} - \lambda\tilde{\xi}_t\Delta\epsilon(x_t^{(k-1)})$; |
| 7    **end** | 7    **end** |
| 8    *Denoising Step*; | 8    *Denoising Step*; |
| 9    $x_{t-1} = \text{Sampler}(x_t^{(K)}, \epsilon^u(x_t^{(K-1)}))$; | 9    $x_{t-1} = \text{Sampler}(x_t^{(K)}, \epsilon^u(x_t^{(K-1)}))$; |
| 10 **end** | 10 **end** |
| 11 **return** $x_0$ | 11 **return** $x_0$ |

## A.3 Z-Sampling [1]

Z-sampling augments conditional guidance by reflective sampling steps: inversion using unconditional noise followed by forward using high-strength guided noise. This procedure is mathematically equivalent to a fixed-point iteration using an backward-forward operator. When integrated with CFG after reflective sampling updates, the process achieves a consistency interval of $t + 1 \to t - 1$ at a cost of 3 NFE per iteration.

Z-Sampling enforces backward-forward consistency via the relation:

$$f_{t\to t+1}^u(x_t) = f_{t\to t+1}^\gamma(x_t) \implies x_t = f_{t\to t+1}^{\gamma^{-1}} \circ f_{t\to t+1}^u(x_t). \tag{20}$$

Here, $f^\gamma$ denotes denoising with noise $\epsilon^u + \gamma(\epsilon^c - \epsilon^u)$. In fact, the equality $f_{t\to t+1}^u(x_t) = f_{t\to t+1}^\gamma(x_t)$ can be viewed as a generalization of $f_{t\to t+1}^u(x_t) = f_{t\to t+1}^c(x_t)$. This stems from the relationship: $\epsilon^u(x_t) = \epsilon^c(x_t)$ implies $\epsilon^u(x_t) = \epsilon^u(x_t) + \gamma[\epsilon^c(x_t) - \epsilon^u(x_t)]$, which establishes that $f_{t\to t+1}^u(x_t) = f_{t\to t+1}^c(x_t)$ can be extended to $f_{t\to t+1}^u(x_t) = f_{t\to t+1}^\gamma(x_t)$.

Approximating the inverse process $f_{t\to t+1}^{\gamma^{-1}} \approx f_{t+1\to t}^\gamma$, we derive:

$$\tilde{x}_t = f_{t+1\to t}^c \circ f_{t\to t+1}^u(x_t). \tag{21}$$

Subsequent application of CFG's calibration step yields:

$$\hat{x}_t = \tilde{x}_t - w\left(\sqrt{1 - \bar{\alpha}_t} - \sqrt{\alpha_t - \bar{\alpha}_t}\right)(\epsilon^c(\tilde{x}_t) - \epsilon^u(\tilde{x}_t)), \tag{22}$$

$$x_{t-1} = \frac{\sqrt{\bar{\alpha}_{t-1}}}{\sqrt{\bar{\alpha}_t}}\hat{x}_t + \left(\sqrt{1 - \bar{\alpha}_{t-1}} - \frac{\sqrt{1 - \bar{\alpha}_t}}{\sqrt{\bar{\alpha}_t}}\right)\epsilon^u(\tilde{x}_t). \tag{23}$$

This constructs a composite fixed-point iteration over the extended interval $t + 1 \to t - 1$.

## A.4 Resampling [20]

Resampling also utilizes backward-forward fixed-point iterations but differs by incorporating a model-free stochastic noise function for the inversion step. Consequently, it maintains the $t + 1 \rightarrow t - 1$ update interval while reducing the computational cost to 2 NFE per iteration.

The stochastic backward step is defined as:

$$n_{t \rightarrow t+1}(x_t) = \sqrt{\alpha_{t+1}} x_t + \sqrt{1 - \alpha_{t+1}} \epsilon, \quad \epsilon \sim \mathcal{N}(0, \mathbf{I}), \tag{24}$$

which leads to the complete update sequence:

$$\tilde{x}_t = f_{t+1 \rightarrow t}^{\gamma} \circ n_{t \rightarrow t+1}^u(x_t), \tag{25}$$

$$\hat{x}_t = \tilde{x}_t - w \left( \sqrt{1 - \bar{\alpha}_t} - \sqrt{\alpha_t - \bar{\alpha}_t} \right) (\epsilon^c(\tilde{x}_t) - \epsilon^u(\tilde{x}_t)), \tag{26}$$

$$x_{t-1} = \frac{\sqrt{\bar{\alpha}_{t-1}}}{\sqrt{\bar{\alpha}_t}} \hat{x}_t + \left( \sqrt{1 - \bar{\alpha}_{t-1}} - \frac{\sqrt{1 - \bar{\alpha}_t}}{\sqrt{\alpha_t}} \right) \epsilon^u(\tilde{x}_t). \tag{27}$$

Although computationally efficient, the stochastic approximation $n_{t \rightarrow t+1}$ introduces semantic distortion compared to deterministic counterparts.

We demonstrate that Resampling can be viewed as an approximation of Z-Sampling when the step size is sufficiently small and $\gamma$ is sufficiently large. First, we express the DDIM forward process $f_{t+1 \rightarrow t}^{\gamma}$ explicitly:

$$f_{t+1 \rightarrow t}^{\gamma}(x_{t+1}) = \frac{1}{\sqrt{\alpha_{t+1}}} x_{t+1} + \left( \sqrt{1 - \bar{\alpha}_t} - \frac{\sqrt{1 - \bar{\alpha}_t}}{\sqrt{\alpha_{t+1}}} \right) \epsilon^{\gamma}(x_{t+1}). \tag{28}$$

For a sufficiently small step size, we approximate $\epsilon(x_{t+1}) \approx \epsilon(x_t)$ and substitute $x_{t+1} = \sqrt{\alpha_{t+1}} x_t + \sqrt{1 - \alpha_{t+1}} \epsilon$ to obtain the fixed-point update form:

$$\begin{aligned}
\hat{x}_t &= \frac{1}{\sqrt{\alpha_{t+1}}} (\sqrt{\alpha_{t+1}} x_t + \sqrt{1 - \alpha_{t+1}} \epsilon) + \left( \sqrt{1 - \bar{\alpha}_t} - \frac{\sqrt{1 - \bar{\alpha}_t}}{\sqrt{\alpha_{t+1}}} \right) \epsilon^{\gamma}(x_{t+1}) \\
&= x_t + m_t \epsilon + n_t [\epsilon^u(x_t) + \gamma(\epsilon^c(x_t) - \epsilon^u(x_t))],
\end{aligned} \tag{29}$$

where $m_t = \sqrt{1 - \alpha_{t+1}}/\sqrt{\alpha_{t+1}}$ and $n_t = \sqrt{1 - \bar{\alpha}_t} - \sqrt{1 - \bar{\alpha}_t}/\sqrt{\alpha_{t+1}}$ are time-dependent constants. Compared to the fixed-point objective $\epsilon^c(x_t) - \epsilon^u(x_t) = 0$ of other operators, the objective for Resampling's fixed-point operator becomes:

$$\epsilon^c(x_t) - \epsilon^u(x_t) = \frac{1}{\gamma} [m_t \epsilon - n_t \epsilon^u(x_t)]. \tag{30}$$

Theoretically, the objective of Resampling aligns with other fixed-point operators when $\gamma$ is sufficiently large. In practice, however, $\gamma$ is not set to an extremely large value to maintain sample quality, which causes Resampling to underperform compared to other operators. Nevertheless, Table 2 shows that Resampling still benefits from increased iteration counts. This empirically confirms that Resampling's fixed-point operator is a functional, albeit suboptimal, choice.

This systematic analysis demonstrates how various guidance methods can be unified under our fixed point framework, as summarized in Table 1 of the main text.

## A.5 Comparison of Design Choices

This section provides a comprehensive analysis of the design choices within our unified fixed point iteration framework, examining both iteration strength schedulers and fixed point operators.

**Iteration strength schedulers.** The schedulers for CFG and CFG++ are given by $\xi_t = \sqrt{1 - \bar{\alpha}_t} - \sqrt{\alpha_t - \bar{\alpha}_t}$ and $\tilde{\xi}_t = \sqrt{1 - \bar{\alpha}_t}$, respectively. Using the $\alpha_t$ setting in DDIM as an example, Figure 6 shows the $\xi_t$ and $\tilde{\xi}_t$ at different timesteps.

It can be observed that, in the early stages where guidance has a greater impact on generation ($t \in [0.67, 1]$), the guidance strength provided by $\xi_t$ decays rapidly, which limits generation quality. In contrast, $\tilde{\xi}_t$ provides more stable iteration strength during these critical early stages, explaining the improved stability of CFG++ over CFG. The mathematical derivation of these schedulers reveals that $\xi_t$ contains an additional term $\sqrt{\alpha_t - \bar{\alpha}_t}$ that causes the rapid decay, while $\tilde{\xi}_t$ maintains a more gradual decrease throughout the diffusion process.

**Fixed point operators.** In FSG, a forward-backward operator with a larger interval $\Delta t$ is selected, as it exhibits a smaller contraction rate, thereby accelerating the convergence of the fixed-point algorithm.

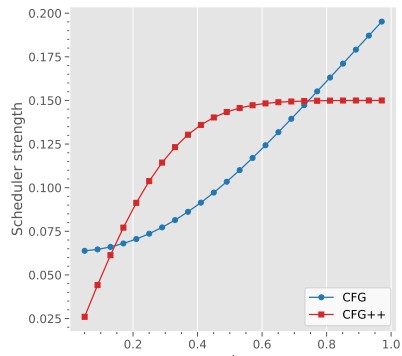

Figure 6: Comparison of iteration strength schedulers for CFG ($\xi_t$) and CFG++ ($\tilde{\xi}_t$) across different timesteps.

Due to the complexity of neural networks, a direct computation of the model's contraction rate is infeasible. Instead, we employ an empirical estimation of the local contraction rate within a neighborhood of $x_t$, defined as a set of points generated by mixing the same $x_0$ with slightly different noises. The local contraction rate for an operator $F$ under the $\ell_2$ norm is measured as:

$$\hat{r} = \mathbb{E}_{x_0 \sim p(x)} \frac{\|F(x_t) - F(x'_t)\|_2^2}{\|x_t - x'_t\|_2^2}, \tag{31}$$

where $x_t, x'_t$ are obtained by mixing different noises with $x_0$.

Table 9: Empirical contraction rates $\hat{r}$ for different operators across denoising timesteps.

| Operators | $t = 0.2$ | $t = 0.4$ | $t = 0.6$ | $t = 0.8$ |
|---|---|---|---|---|
| $\mathrm{id} - w_t \Delta\epsilon$ (CFG, CFG++) | 1.00 | 1.00 | 1.00 | 1.00 |
| $f^{\gamma}_{t+dt \to t} \circ f^{u}_{t \to t+dt}$ (Z-sampling) | 1.04 | 0.99 | 0.97 | 0.99 |
| $f^{\gamma}_{t+dt \to t} \circ n^{u}_{t \to t+dt}$ (Resampling) | 0.89 | 0.97 | 1.03 | 1.07 |
| $f^{u}_{t/2 \to t} \circ f^{\gamma}_{t \to t/2}$ | 1.03 | 0.95 | 0.96 | 0.98 |
| $f^{u}_{t/4 \to t} \circ f^{\gamma}_{t \to t/4}$ | 0.61 | 0.91 | 0.88 | 0.91 |
| $f^{u}_{0 \to t} \circ f^{\gamma}_{t \to 0}$ | 0.62 | 0.70 | 0.75 | 0.79 |

Notably, long-interval operators (e.g., $f^{u}_{0 \to t} \circ f^{\gamma}_{t \to 0}$) demonstrate lower contraction rates ($\hat{r} < 1$) compared to short-interval operators such as Z-sampling or Resampling. To balance convergence speed and operator complexity, we select an intermediate interval $\Delta t$ ($dt < \Delta t < t$) for the operator $f^{u}_{t-\Delta t \to t} \circ f^{\gamma}_{t \to t-\Delta t}$.

The contraction rate of an operator depends on the choice of metric. While operators in CFG/CFG++ are non-contracting under the $\ell_2$ norm, they may exhibit contraction in other carefully designed metrics. Empirically, as shown in Table 2, these methods also benefit from additional iterations, supporting the fixed-point framework.

# B Discussion of Golden Path

The concept of the golden path, wherein latent representations yield consistent outputs under both conditional and unconditional generation, is central to our framework. This section provides empirical evidence supporting this theoretical foundation and demonstrates its practical implications. We construct matched and unmatched samples $x_t$ (on and off the golden path, respectively) by design.

- **Match case:** An image $x^c$ is generated conditioned on a prompt $c$. We then obtain $x_t$ via DDIM inversion [27] using unconditional noise $\epsilon^u(x_t)$, i.e., $x_t = f^{u}_{0 \to t}(x^c)$. These $x_t$ exhibit consistency between conditional and unconditional generation paths, satisfying $x^c \approx f^{u}_{t \to 0}(x_t) \approx f^{c}_{t \to 0}(x_t)$.

- **Mismatch case:** For each matched sample, we assign a different prompt $c' \neq c$ to create a mismatch, where $x^c \approx f^u_{t \to 0}(x_t) \neq f^{c'}_{t \to 0}(x_t)$.

Using these $x_t$ as starting points, we evaluate generation performance under matched and mismatched prompt conditions using CFG [15]. As shown in Table 10, when $f^{c/u}t \to 0(x_t)$ is mismatched, both the alignment and quality of the generated images decrease significantly. Intuitively, when $f^u t \to 0(x_t)$ aligns poorly with condition $c$, a larger proportion of the components in $x_t$ become unsuitable for generating content consistent with $c$. This forces the diffusion model to make more substantial corrections within the limited $t \to 0$ interval, ultimately

Table 10: Performance from $x_t$ under matched ($c$) and mismatched ($c' \neq c$) conditions.

| $t$ | IR Match | IR Mismatch | HPSv2 Match | HPSv2 Mismatch |
|-----|------|----------|-------|----------|
| 0.8 | 97.09 | -69.96 | 28.85 | 25.02 |
| 0.6 | 89.45 | -171.16 | 28.73 | 22.84 |
| 0.4 | 84.42 | -196.38 | 28.53 | 21.86 |
| 0.2 | 81.43 | -201.60 | 28.47 | 21.53 |

compromising generation quality and prompt adherence. These results empirically validate our theoretical framework and motivate the use of fixed-point iterations to achieve golden path consistency.

## C Proofs

We now present the theoretical foundations for FSG, summarized in Theorem 1. We first clarify the notations and state the key assumptions. Under mild conditions, we provide theoretical guidance for allocating the total number of fixed-point iterations $N$. Recall that we partition the diffusion process into $M$ intervals $t_i \to t_i - \Delta t_i$ for $i = 1, \ldots, M$, and perform $K_i$ fixed-point iterations on each interval using a forward-backward operator $F_i := F_{t_i \to t_i - \Delta t_i}$. Specifically, for each iteration $j = 1, \ldots, K_i$ for FSG:

1. Starting from $x_{t_i}^{(j-1)}$, apply the conditionally-guided ODE solver $f^\gamma_{t_i \to t_i - \Delta t_i}$ to obtain $x_{t_i - \Delta t_i}^{(j-1)}$.

2. Then, use the unconditionally-guided reverse ODE solver $f^\gamma_{t_i - \Delta t_i \to t_i}$ to map back and update $x_{t_i}^{(j)}$.

This forward-backward procedure iteratively calibrates $x_{t_i}$ by leveraging both conditional and unconditional guidance over the interval, promoting consistency between the two trajectories. For other methods such as CFG and CFG++, the operator $\mathcal{F}_i$ can be defined as described in Section A (Appendix A). Our analysis remains applicable under these definitions.

For the clarity of theoretical analysis, we consider a fixed-length interval partition, setting each interval to size $W$ so that the total number of timesteps $T$ is divided into $M = T/W$ intervals. We assume $M$ divides $N$, and allocate $K = N/M$ fixed-point iterations to calibrate each $x_{t_i}$, where $t_i = iW$ for $i = 1, \ldots, M$. This leads to the following procedure to produce the trajectory $\hat{x}_t$ from given $x_T$:

$$\hat{x}_t = \begin{cases} F^K_{t \to t - W}(x_t), & \text{if } t \bmod W = 0 \\ x_t, & \text{otherwise} \end{cases}, \quad x_{t-1} = f^u_{t \to t-1}(\hat{x}_t), \quad t = 1 \ldots T. \quad (32)$$

Let $\varepsilon^u(x, t)$ and $\varepsilon^c(x, t)$ denote the unconditional and conditional noise predictions, respectively. The domain regarding $x$ for all the functions, including $\varepsilon^{u/c}(x, t)$, $f^{u/c}_{a \to b}(x)$ and $F_{a \to b}(x)$, etc., is assumed to be $\mathbb{R}^d$ (unless otherwise specified). And we use $\|\cdot\|$ to denote the Euclidean norm in $\mathbb{R}^d$.

Key assumptions are as follows:

1. **Boundedness:** The latent variables and noise predictions are bounded, i.e., $\|\hat{x}_t\|, \|\epsilon^{c/u}(\hat{x}_t)\| \leq B$ for some constant $B$ (assumption 1).

2. **Smoothness:** The noise functions $\epsilon^{c/u}(\cdot)$ are Lipschitz continuous with a smoothness constant $L$ (assumption 2). Additionally, the ODE dynamics derived from these noise functions are smooth (assumption 3).

3. **Contraction:** The fixed-point operator $F_i$ is a contraction with a rate bounded by $r \in (0, 1)$ (assumption 4).

Building on the assumptions, we establish a framework for optimizing the allocation of fixed-point iterations. For the procedure described in (32), we analyze the allocation of iterations across intervals $[(i-1)W, iW]$ for $i \in [1, M]$, where the fixed-point operator $F_{iW \to (i-1)W}$ is applied. The sampling ratio is defined as:

$$\beta := \frac{1}{M} = \frac{K}{N} = \frac{W}{T}, \tag{33}$$

where $M$ is the number of intervals, $K$ is the number of iterations per interval, $N$ is the total number of iterations, and $T$ is the total number of timesteps. We derive an upper bound to guide the optimal choice of $\beta$.

**Theorem** (detailed version of Theorem 1). *Assume that Assumptions 1 to 4 hold. Let $\hat{x}_t$ denote the trajectory produced as in* (32), *with $\beta = \frac{1}{M} \in (0, 1)$. There exists an upper bound function:*

$$g(\beta, L, r) = \left(\Theta(1) + \Theta(L^2)\right) r^{2\beta N} + 2L^2\beta^2, \tag{34}$$

*such that:*

$$\mathcal{L} = \sum_{t=1}^{T} \|\varepsilon^c(\hat{x}_t, t) - \varepsilon^u(\hat{x}_t, t)\|^2 \leq B^2 T g(\beta, L, r). \tag{35}$$

The proof is deferred to the end of this section.

Here, $O(f(L))$ denotes an upper bound up to a constant, i.e., $\leq C f(L)$ for some $C > 0$; $\Theta(f(L))$ denotes both upper and lower bounds up to constants, i.e., $cf(L) \leq \cdot \leq Cf(L)$ for some $c, C > 0$. The assumption 1 provides a trivial bound $\mathcal{L} \leq 4B^2T$, while large $L$ may lead to $g(\beta, L, r) > 4$, so our bound is more meaningful for smooth noise function with smaller $L$. For $L = O(1)$, the bound simplifies to:

$$g(\beta, L, r) = \Theta(r^{2\beta N}) + 2L^2\beta^2. \tag{36}$$

**Optimal $\beta$ for minimizing $g(\beta, L, r)$:** For fixed $N$, $L$, and $r$, the optimal $\beta^*$ is obtained by solving:

$$\frac{d}{d\beta}g(\beta, L, r) = 4L^2\beta - \Theta\left(2N \ln(1/r)(1/r)^{-2\beta N}\right) = 0, \tag{37}$$

$$L^2\beta = \Theta(N \ln(1/r)(1/r)^{-2\beta N}).$$

Here, $L^2\beta$ increases linearly with $\beta$, while the term $N \ln(1/r)(1/r)^{-2\beta N}$ decays exponentially, guaranteeing a unique minimizer $\beta^*$. Importantly, as the noise smoothness $L$ decreases, the optimal $\beta^*$ increases, suggesting that smoother noise allows for fewer, longer intervals (i.e., larger $W$ and $K$). Conversely, as the total iteration budget $N$ grows, $\beta^*$ approaches zero, recovering the short-interval setting ($M = T$) as a limiting case.

Next, we formalize the boundedness assumptions for images and noise, which are standard in diffusion model analysis:

**Assumption 1** (Boundedness). *There exists $B > 0$ such that for all $t$ and all $x \in \mathcal{M}_t$,*

$$\|x\| \leq B, \quad \|\varepsilon^u(x, t)\| \leq B. \tag{38}$$

For images with entries in $[0, 1]$ and size $d_1 \times d_2$, we may take $d = d_1 d_2$ and $B = \sqrt{d_1 d_2}$.

We next formalize the smoothness properties of the noise function, introducing a constant $L = O(1)$ that is independent of the data dimension $d$ and the number of diffusion steps $T$.

**Assumption 2** (Smoothness of Noise). *There exists a smoothness constant $L \in (0, C)$, such that the following properties hold for any initial point $x$ and time interval $[a, b]$:*

1. ***Smoothness at a fixed time:** For any $t \in [a, b]$ and any pair of points $x_1, x_2$, the noise function satisfies:*

$$\|\varepsilon^{u/c}(x_1, t) - \varepsilon^{u/c}(x_2, t)\| \leq L\|x_1 - x_2\|. \tag{39}$$

2. **Smoothness over varying time:** *Let $x_t^{u/c} = f_{a \to b}^{u/c}(x)$ for $t \in [a, b]$. Then, for any $t_1, t_2 \in [a, b]$, the noise function satisfies:*

$$\|\varepsilon^{u/c}\left(x_{t_1}^u, t_1\right) - \varepsilon^{u/c}\left(x_{t_2}^u, t_2\right)\| \leq LB\frac{|t_1 - t_2|}{T},$$

$$\|\varepsilon^{u/c}\left(x_{t_1}^c, t_1\right) - \varepsilon^{u/c}\left(x_{t_2}^c, t_2\right)\| \leq LB\frac{|t_1 - t_2|}{T}.$$

Building on the smoothness of noise (assumption 2), we analyze the properties of $f_{a \to b}^{u/c}(x)$. Recall that $f_{a \to b}^{u/c}(x)$ represents an ODE evolving from time $a$ to $b$, initialized at point $x$. The ODE is defined as:

$$\frac{d}{dt}\left(\frac{x_t^{u/c}}{\sqrt{\bar{\alpha}_t}}\right) = \frac{d}{dt}\left(\sqrt{\frac{1 - \bar{\alpha}_t}{\bar{\alpha}_t}}\right) \varepsilon_\theta^{u/c}\left(x_t^{u/c}, t\right), \tag{40}$$

where $\theta$ denotes the parameters of the neural network, which we omit for simplicity in subsequent discussions. This ODE yields the following explicit form for the time derivative of $x_t^{u/c}$:

$$\frac{dx_t^{u/c}}{dt} = \mu_t x_t^{u/c} + \lambda_t \varepsilon^{u/c}\left(x_t^{u/c}, t\right), \tag{41}$$

where

$$\lambda_{t_0} := \sqrt{\bar{\alpha}_t} \frac{d}{dt}\left(\sqrt{\frac{1 - \bar{\alpha}_t}{\bar{\alpha}_t}}\right)\Big|_{t=t_0}, \quad \mu_{t_0} := -\sqrt{\bar{\alpha}_t} \frac{d}{dt}\left(\frac{1}{\sqrt{\bar{\alpha}_t}}\right)\Big|_{t=t_0}. \tag{42}$$

**Assumption 3.** *Let $L$ be the smoothness constant in assumption 2. There exist constants $C > 1$ and $c \in (0, 1)$ such that:*

1. **Smooth Dependence on Initial Error:**

$$\|f_{a \to b}^{u/c}(x) - f_{a \to b}^{u/c}(y)\| \leq C\ell_{a,b,L}\|x - y\|, \tag{43}$$

   *where*

$$\ell_{a,b,L} = (|\lambda_a|L + |\mu_a|)|b - a| + 1. \tag{44}$$

2. **Noise Control:**

$$\|f_{a \to b}^c(x) - f_{a \to b}^u(x)\| \geq c\lambda_a\|\varepsilon^c(x, a) - \varepsilon^u(x, a)\||b - a|. \tag{45}$$

**Motivation:** This assumption is motivated by a first-order Taylor expansion of $f_{a \to b}^{u/c}(x/y)$.

- For smoothness on intial error:

$$f_{a \to b}^{u/c}(x) - f_{a \to b}^{u/c}(y) \approx \frac{d}{dt}\left(f_{a \to t}^{u/c}(x) - f_{a \to t}^{u/c}(y)\right)\Big|_{t=a}(b - a) + (x - y)$$

$$= \left(\mu_a(x - y) + \lambda_a\left(\varepsilon^{u/c}(x, a) - \varepsilon^{u/c}(y, a)\right)\right)(b - a) + (x - y),$$

  $\|\varepsilon^{u/c}(x, a) - \varepsilon^{u/c}(y, a)\| \leq L\|x - y\|$ from assumption 2,

  $\|f_{a \to b}^{u/c}(x) - f_{a \to b}^{u/c}(y)\| \leq C\ell_{a,b,L}\|x - y\|,$

  where $\ell_{a,b,L} := (|\lambda_a|L + |\mu_a|)|b - a| + 1$ captures the effect of the smoothness of noise, the length of time interval $[a, b]$ the and dynamics of $\bar{\alpha}_t$ in it. $C > 1$ is a constant that compensates for the error in the Taylor expansion.

- For noise control:

$$f_{a \to b}^u(x) - f_{a \to b}^c(x) \approx \frac{d}{dt}\left(f_{a \to t}^u(x) - f_{a \to t}^c(x)\right)\Big|_{t=a}(b - a)$$

$$= \lambda_a\left(\varepsilon^c(x, a) - \varepsilon^u(x, a)\right)(b - a),$$

  $\|f_{a \to b}^u(x) - f_{a \to b}^c(y)\| \geq c|\lambda_a|\|\varepsilon^c(x, a) - \varepsilon^u(x, a)\||b - a|,$

  where $c \in (0, 1)$ compensates for the error in the Taylor expansion.

**Assumption 4** (Contraction and Fixed Point Existence). *Let $r \in (0,1)$. For any time interval $[a,b]$, the fixed point operator $F_{a \to b}$ satisfies the following:*

1. ***Fixed point equivalence:*** *There exists a fixed point $x^*$ of $F_{a \to b}$ such that:*
$$F_{a \to b}(x^*) = x^* \quad \Leftrightarrow \quad f^c_{a \to b}(x^*) = f^u_{a \to b}(x^*). \tag{46}$$

2. ***Contraction:*** *For all $x, y$:*
$$\|F_{a \to b}(x) - F_{a \to b}(y)\| \le r \|x - y\|. \tag{47}$$

As an example, the main text considers the fixed point operator $F_{a \to b}(x) = f^{u(-1)}_{a \to b} \circ f^c_{a \to b}(x)$. For brevity, we write $F(x)$ when the interval $[a,b]$ is clear from context.

**Lemma 1.** *Assume that Assumptions 1 to 4 hold. For any given $x$, the following inequality holds:*
$$\|f^c_{a \to b}\left(F^{(k)}(x)\right) - f^u_{a \to b}\left(F^{(k)}(x)\right)\| \le 2C\ell_{a,b,L} r^k \|x - x^*\| \le 4C\ell_{a,b,L} r^k B. \tag{48}$$

*Proof.* We start by expanding the difference:
$$\|f^c_{a \to b}\left(F^{(k)}(x)\right) - f^u_{a \to b}\left(F^{(k)}(x)\right)\|$$
$$= \|f^c_{a \to b}\left(F^{(k)}(x)\right) - f^u_{a \to b}\left(F^{(k)}(x)\right) - \left(f^c_{a \to b}\left(F^{(k)}(x^*)\right) - f^u_{a \to b}\left(F^{(k)}(x^*)\right)\right)\|$$
$$\le \|f^c_{a \to b}\left(F^{(k)}(x)\right) - f^c_{a \to b}\left(F^{(k)}(x^*)\right)\| + \|f^u_{a \to b}\left(F^{(k)}(x)\right) - f^u_{a \to b}\left(F^{(k)}(x^*)\right)\|.$$

Using the smoothness property of $f^c_{a \to b}(\cdot)$ and $f^u_{a \to b}(\cdot)$ from assumption 3, we have:
$$\|f^c_{a \to b}\left(F^{(k)}(x)\right) - f^c_{a \to b}\left(F^{(k)}(x^*)\right)\| \le C\ell_{a,b,L}\|F^{(k)}(x) - F^{(k)}(x^*)\|, \tag{49}$$
$$\|f^u_{a \to b}\left(F^{(k)}(x)\right) - f^u_{a \to b}\left(F^{(k)}(x^*)\right)\| \le C\ell_{a,b,L}\|F^{(k)}(x) - F^{(k)}(x^*)\|. \tag{50}$$

Combining these, we get:
$$\|f^c_{a \to b}\left(F^{(k)}(x)\right) - f^u_{a \to b}\left(F^{(k)}(x)\right)\| \le 2C\ell_{a,b,L}\|F^{(k)}(x) - F^{(k)}(x^*)\|. \tag{51}$$

Next, applying the contraction property of $F$ from assumption 4, we have:
$$\|F^{(k)}(x) - F^{(k)}(x^*)\| \le r^k \|x - x^*\|. \tag{52}$$

Substituting this into the inequality, we obtain:
$$\|f^c_{a \to b}\left(F^{(k)}(x)\right) - f^u_{a \to b}\left(F^{(k)}(x)\right)\| \le 2\ell_{a,b,L} r^k \|x - x^*\|. \tag{53}$$

Finally, using the boundedness assumption from assumption 1, $\|x - x^*\| \le 2B$, we conclude:
$$\|f^c_{a \to b}\left(F^{(k)}(x)\right) - f^u_{a \to b}\left(F^{(k)}(x)\right)\| \le 4\ell_{a,b,L} r^k B. \tag{54}$$
$\square$

With these assumptions and lemmas established, we are now ready to prove the main theorem.

*Proof.* We analyze the error in the first interval, as the analysis for other intervals is similar. Let $F$ bes the simplified notation for $F_{W \to 0}$. The error can be bounded as follows:
$$\sum_{t=1}^{W} \|\varepsilon^c(\hat{x}_t, t) - \varepsilon^u(\hat{x}_t, t)\|^2$$
$$= \|\varepsilon^c(\hat{x}_W, W) - \varepsilon^u(\hat{x}_W, W)\|^2 + \sum_{t=1}^{W-1} \|\varepsilon^c(\hat{x}_t, t) - \varepsilon^u(\hat{x}_t, t)\|^2 \tag{55}$$
$$\overset{(a)}{\le} (1 + 3(W-1))\|\varepsilon^c(\hat{x}_W, W) - \varepsilon^u(\hat{x}_W, W)\|^2$$
$$+ 3\sum_{t=1}^{W-1} \left(\|\varepsilon^c(\hat{x}_t, t) - \varepsilon^c(\hat{x}_W, W)\|^2 + \|\varepsilon^u(\hat{x}_t, t) - \varepsilon^u(\hat{x}_W, W)\|^2\right),$$

where $(a)$ follows from $\|a + b + c\|^2 \leq 3(\|a\|^2 + \|b\|^2 + \|c\|^2)$.

Using the bounds in assumption 2 and assumption 3, we have:

$$
\sum_{t=1}^{W}\|\varepsilon^c\left(\hat{x}_t, t\right) - \varepsilon^u\left(\hat{x}_t, t\right)\|^2
$$

$$
\leq (3W - 2)\|f_{W \to 0}^c\left(F^{(K)}\left(x_W\right)\right) - f_{W \to 0}^u\left(F^{(K)}\left(x_W\right)\right)\|^2 \frac{1}{c^2 \lambda_W^2 W^2}
$$

$$
+ 6\frac{B^2}{T^2}L^2 \sum_{i=1}^{W-1}(W - i)^2
\tag{56}
$$

$$
\overset{Lemma\ 1}{\leq} (4\ell_{W,0,L}r^K B)^2 \frac{C^2(3W - 2)}{c^2 \lambda_W^2 W^2} + 6\frac{B^2}{T^2}L^2 \frac{(W - 1)W(2W - 1)}{6}.
$$

Now, summing over all intervals:

$$
\sum_{t=1}^{T}\|\varepsilon^c\left(\hat{x}_t, t\right) - \varepsilon^u\left(\hat{x}_t, t\right)\|^2
$$

$$
= \sum_{i=1}^{M}\sum_{t=1}^{W}\|\varepsilon^c\left(\hat{x}_{iW-W+t}, iW - W + t\right) - \varepsilon^u\left(\hat{x}_{iW-W+t}, iW - W + t\right)\|^2
$$

$$
\leq \sum_{i=1}^{M}\left((4\ell_{iW,(i-1)W,L}r^K B)^2 \frac{C^2(3W - 2)}{c^2 \lambda_{iW}^2 W^2}\right.
$$

$$
\left. + 6\frac{B^2}{T^2}L^2 \frac{(W - 1)W(2W - 1)}{6}\right)
\tag{57}
$$

$$
\overset{MW=T}{=} B^2\left(16 \sum_{i=1}^{T/W}\ell_{iW,(i-1)W,L}^2 \frac{C^2(3W - 2)}{c^2 \lambda_{iW}^2 W^2}r^{2K}\right.
$$

$$
\left. + 6\frac{L^2(W - 1)(2W - 1)}{6T}\right).
$$

Expanding $\ell_{iW,(i-1)W,L}$, we obtain:

$$
\sum_{t=1}^{T}\|\varepsilon^c\left(\hat{x}_t, t\right) - \varepsilon^u\left(\hat{x}_t, t\right)\|^2
$$

$$
\leq 16B^2 \sum_{i=1}^{T/W}((|\lambda_{iW}|L + |\mu_{iW}|)W + 1)^2 \frac{C^2(3W - 2)}{c^2 \lambda_{iW}^2 W^2}r^{2K}
$$

$$
+ 6B^2 \frac{L^2(W - 1)(2W - 1)}{6T}
$$

$$
\overset{(W-1)(2W-1)\leq 2\beta^2 T^2}{\leq} 16B^2 \sum_{i=1}^{T/W}((|\lambda_{iW}|L + |\mu_{iW}|)W + 1)^2 \frac{C^2(3W - 2)}{c^2 \lambda_{iW}^2 W^2}r^{2K} + 6B^2 \frac{L^2 2\beta^2 T^2}{6T}
$$

$$
\leq B^2 \times 16 \sum_{i=1}^{T/W}((|\lambda_{iW}|L + |\mu_{iW}|)W + 1)^2 \frac{C^2(3W - 2)}{c^2 \lambda_{iW}^2 W^2}r^{2K} + B^2 T(2L^2 \beta^2).
$$

$$
\tag{58}
$$

Observe that $|\lambda_{iW}|, |\mu_{iW}|, i \in [M]$ are constants, and we define

$$
C_1 = 16 \sum_{i=1}^{T/W}((|\lambda_{iW}|L + |\mu_{iW}|)W + 1)^2 \frac{C^2(3W - 2)}{c^2 \lambda_{iW}^2 W^2}/T = \Theta(L^2) + \Theta(1).
$$

Finally, grouping terms:

$$\mathcal{L} = \sum_{t=1}^{T} \|\varepsilon^c(\hat{x}_t, t) - \varepsilon^u(\hat{x}_t, t)\|^2 \leq B^2 T g(\beta, L, r), \tag{59}$$

where

$$g(\beta, L, r) = \left(\Theta(1) + \Theta(L^2)\right) r^{2\beta N} + 2L^2 \beta^2.$$

$\square$

## D  Experimental Details

### D.1  Experimental Setup

In this section, we demonstrate the specific setup of the experiment, including the dataset, metrics, and hyperparameter settings for different methods.

**Datasets.**  The evaluation leverages four datasets to assess text-to-image models.

1. **Pick-a-Pic** [17] collects real-world user preferences from a text-to-image web app. For each prompt, users compare two generated images and select their preferred option (or mark a tie). We use the first 100 prompts from this dataset to test model performance.

2. **DrawBench** [25] is a comprehensive and challenging benchmark with ∼200 prompts spanning 11 categories like color, counting, and text rendering. These prompts test how well models handle complex or ambiguous descriptions.

3. **GenEval** [10] evaluates whether generated images correctly follow object-focused instructions. Its 553 prompts cover object placement, quantity, color, leveraging object detection models to evaluate text-to-image models on a variety of generation tasks.

4. **PartiPrompts** [32] contains 1,600+ diverse prompts covering creative, technical, and abstract concepts. We randomly pick 100 prompts to assess how models balance language understanding and visual creativity.

**Metrics.**  Four metrics are employed to quantify image quality and alignment.

1. **Aesthetic Score (AES)** [26] quantifies assigning scores (often 1–10) of visual quality by analyzing contrast, composition, color harmony, and detail richness. AES reflects human aesthetic preferences to help refine image generation.

2. **ImageReward (IR)** [30] is a reward model trained on 137,000 expert comparisons using rating/ranking methods, which can be integrated with reinforcement learning to enhance output alignment with human preferences.

3. **Human Preference Score v2 (HPSv2)** [29] is fine-tuned from CLIP based on Human Preference Dataset v2 with 798,090 human preference choices. HPSv2 can predicts authentic human perceptions of beauty and style.

4. **CLIPScore** [12] measures text-image alignment via CLIP embedding cosine similarity. This metric is essential in text-to-image synthesis, ensuring that generated visuals maintain semantic alignment.

**Hyper-parameters.**  For Classifier-Free Guidance (CFG), we adopted a guidance strength $w = 5.5$, while CFG++ used $\lambda = 0.6$. Both methods utilized 50 inference steps. In Z-Sampling, forward guidance strength was set to 5.5, and reverse guidance strength to 0. Following the setting in [1], reflective sampling was applied during first 12/25 steps for NFE=50, 25/50 steps for NFE=100, and all 50 steps for NFE=150. For Resampling, configurations varied by NFE: at NFE=50, 25 inference steps with one resample per step; for NFE=100/150, 50 steps with 1 or 2 resamples per step, respectively.

For our Foresight Guidance (FSG) method, we set $\lambda = 1.0$ for NFE=50/100 and $\lambda = 0.7$ for NFE=150. We allocate fixed-point iterations $(t_i, \Delta t_i, K_i)$ using a stage-wise strategy that prioritizes early timesteps:

Table 11: Performance under different hyperparameter settings. Dataset: Pick-a-Pic; NFE: 100.

| Setting | IR↑ | HPSv2↑ | AES↑ | CLIP↑ |
|---|---|---|---|---|
| 3:2:1 (Default) | 102.82 | 29.05 | 6.66 | 34.30 |
| 1:1:1 | 99.13 | 28.95 | 6.67 | 34.14 |
| 1:2:3 | 93.23 | 28.70 | 6.63 | 33.75 |
| Intra-stage perturbation | 102.29 ($\pm$ 1.01) | 28.72 ($\pm$ 0.22) | 6.59 ($\pm$ 0.04) | 34.33 ($\pm$ 0.17) |
| Interval perturbation | 102.17 ($\pm$ 0.79) | 29.04 ($\pm$ 0.02) | 6.67 ($\pm$ 0.01) | 34.28 ($\pm$ 0.04) |

Table 12: The quantitative results on PartiPrompts Dataset with the SDXL model and NFE = 50, 100, 150 (↑ denotes higher is better, best results under the same NFE are bolded).

| Method | NFE | IR↑ | HPSv2↑ | AES↑ | CLIP↑ |
|---|---|---|---|---|---|
| CFG | 50 | 99.13 | 28.72 | 6.41 | 33.01 |
| CFG++ | 50 | 106.80 | 28.96 | **6.42** | 33.20 |
| Z-Sampling | 50 | 114.74 | 29.04 | 6.32 | 33.41 |
| Resampling | 50 | 96.73 | 28.73 | 6.17 | 33.35 |
| FSG (ours) | 50 | **117.65** | **29.20** | 6.30 | **33.53** |
| CFG×2 | 100 | 117.84 | 29.25 | 6.33 | 33.42 |
| CFG++×2 | 100 | 115.35 | 29.33 | 6.26 | 33.46 |
| Z-Sampling | 100 | 118.39 | 29.22 | **6.34** | **33.47** |
| Resampling | 100 | 112.72 | 29.23 | 6.31 | 33.20 |
| FSG (ours) | 100 | **121.11** | **29.38** | 6.29 | 33.46 |
| CFG×3 | 150 | 117.00 | 29.37 | 6.22 | 33.40 |
| CFG++×3 | 150 | 117.02 | 29.29 | 6.22 | **33.41** |
| Z-Sampling | 150 | 120.20 | 29.35 | **6.35** | 33.34 |
| Resampling | 150 | 113.62 | 29.16 | 6.24 | 33.16 |
| FSG (ours) | 150 | **123.28** | **29.40** | 6.34 | **33.41** |

- **NFE=50** (limited budget): We employ 40 inference steps and concentrate iterations on early stages. Fixed-point iterations are performed at timesteps $t \in \{1.0, 0.875, 0.625\}$ with interval size $\Delta t = 0.125$, executing $K = 2, 2, 1$ iterations respectively.

- **NFE=100** (moderate budget): We adopt 50 inference steps with a 3:2:1 allocation ratio across early, middle, and late stages. Specifically: (i) Early stage ($t \in \{0.68, 0.72, \cdots, 1.0\}$): 24 NFEs with $\Delta t = 0.06$, applying 2 iterations at $t \in \{1.0, 0.92, 0.8\}$ and 1 iteration at remaining timesteps; (ii) Middle stage ($t \in \{0.36, 0.40, \cdots, 0.64\}$): 16 NFEs with $\Delta t = 0.04$; (iii) Late stage ($t \in \{0.08, 0.14, \cdots, 0.32\}$): 10 NFEs with $\Delta t = 0.02$.

- **NFE=150** (ample budget): With sufficient computational resources, we augment the NFE=100 configuration by adding supplementary fixed-point iterations at intermediate timesteps $\{0.02, 0.06, \ldots, 0.98\}$ with interval $\Delta t = 0.02$ to enhance image detail.

Despite the apparent complexity of the hyperparameters $\{(t_i, \Delta t_i, K_i)\}$, they are relatively straightforward to configure and do not require extensive fine-tuning. As demonstrated in Table 11, FSG maintains stable performance under random perturbations to timestep positions ($\pm 0.02$) or interval lengths ($\times 0.7$-$1.3$), provided the overall stage-wise allocation ratio (e.g., 3:2:1) is preserved.

## D.2 Experiment Results in PartiPrompts Dataset

In this section, we present comparative experiments between our method and various baselines on the PartiPrompts dataset, aiming to mitigate dataset-induced bias and further validate the effectiveness of our approach.

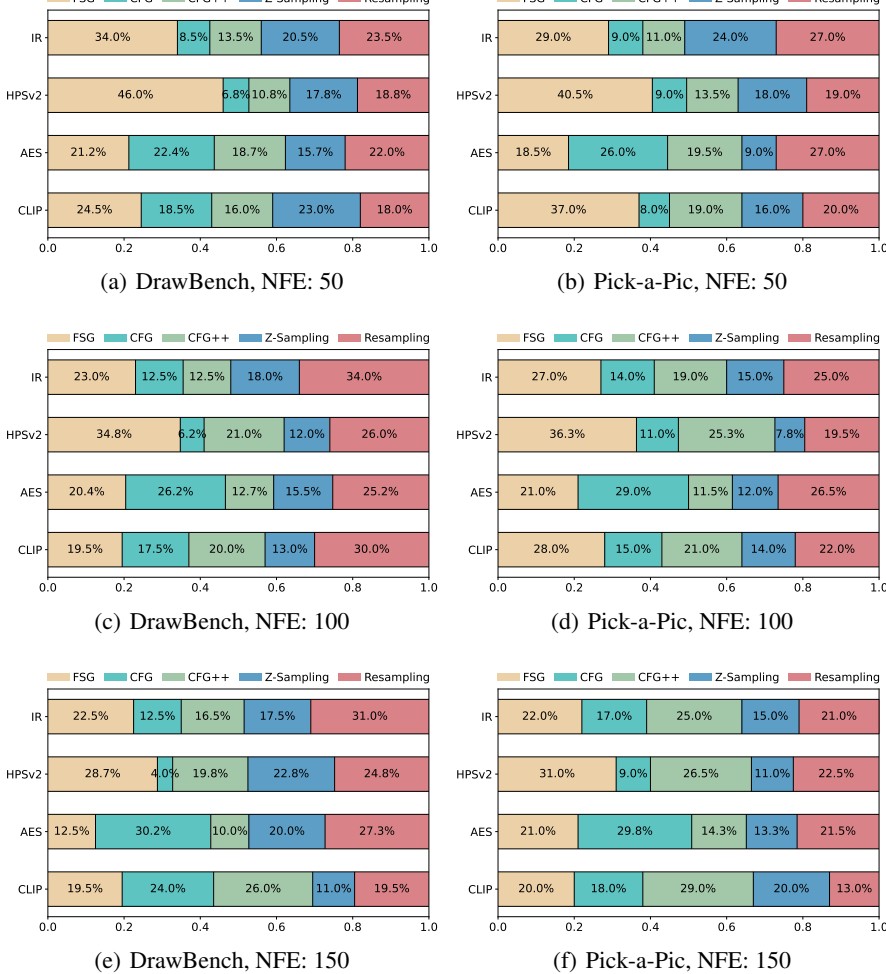

Figure 7: Proportion of samples that outperform the other four methods (Top-1 rate).

As shown in Table 12, our method FSG consistently achieves state-of-the-art performance across IR and HPSv2 metrics. For the CLIP metric, FSG outperforms all other baselines at NFE = 50 and NFE = 150, with only a marginal difference (-0.01) against the best performance at NFE = 100. Notably, FSG demonstrates substantial improvements over CFG in the IR metric across all NFE levels (50, 100, and 150), with performance gains of +18.52, +3.27, and +6.28 respectively. Compared to the second-best method, we achieve consistent improvements of +2.91, +2.72, and +3.08, demonstrating superior image generation quality. The strong performance on both HPSv2 and CLIP metrics further confirms that our method maintains excellent alignment while producing high-quality outputs.

These results collectively demonstrate that our fixed-point iteration strategy enables more balanced sub-problem decoupling, leading to better convergence behavior and improved performance in both image quality and text alignment. These findings align with the results obtained on the DreamBench and Pick-a-Pic datasets, as presented in Table 2 of the main text.

## D.3 Additional Results on Top-1 Rate

In this section, we present the Top-1 rate of different methods on SDXL. As evidenced by the Top-1 rate in Figure 7, FSG exhibits a significant advantage at NFE=50, particularly in HPSv2 (46% on DrawBench, 40.5% on Pick-a-Pic) and IR (34% on DrawBench, 29% on Pick-a-Pic). Even as NFE increases, FSG retains superiority in HPSv2, suggesting that long-interval guidance effectively strengthens alignment with human preferences.

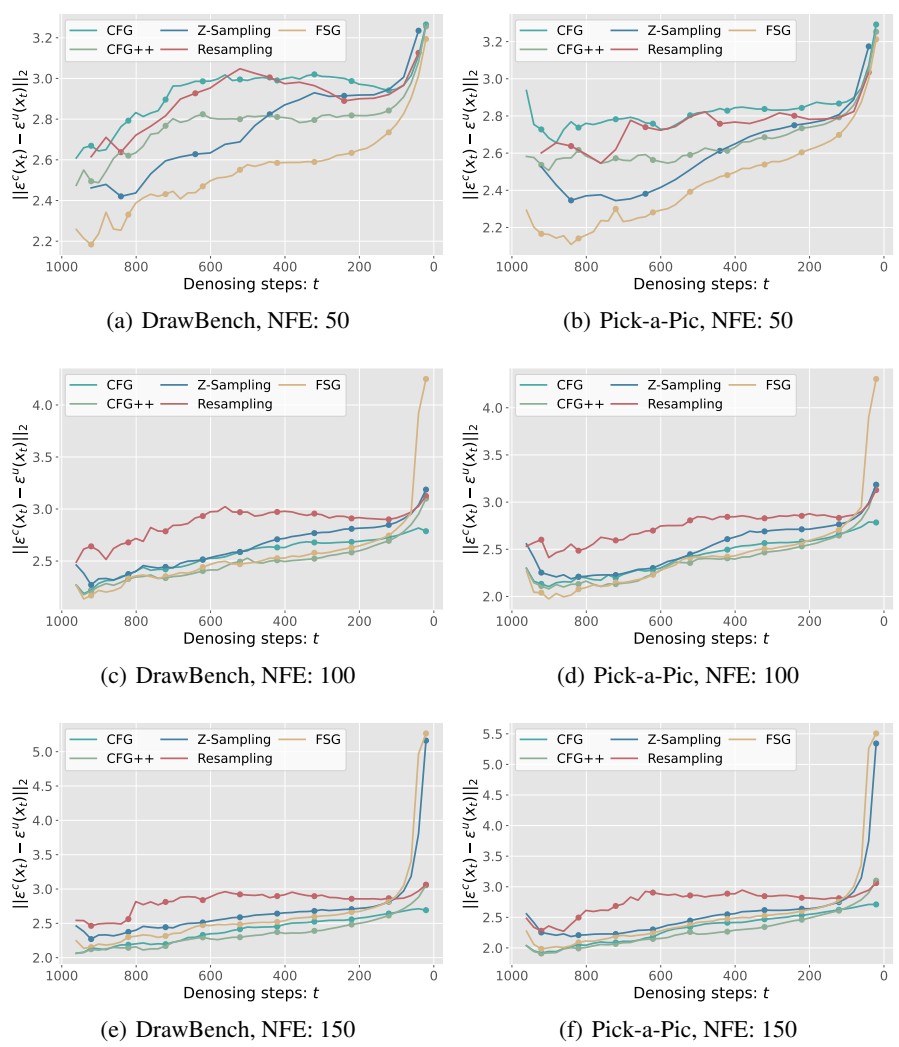

Figure 8: Prediction gap during denoising, indicating the effectiveness of the fixed point iteration.

A notable observation is the divergent performance characteristics of fixed-point iteration variants. For example, at NFE=150, CFG enhances AES metrics, while Resampling improves IR of specific samples, likely attributable to its stochastic nature. These findings highlight the potential for adaptive frameworks that exploit the unique properties of fixed-point operators to address diverse objectives, presenting a promising direction for future research.

### D.4 Additional Results on Prediction Gap

In this section, we analyze the noise prediction gap $\|\epsilon^u(x_t) - \epsilon^c(x_t)\|_2^2$ during the denoising process of different methods on SDXL. We select the prediction gap as a key metric since it provides insight into score estimation accuracy along the generation trajectory. We hypothesize that models achieve optimal score [28] (noise) estimation when $\epsilon^c(x_t) - \epsilon^u(x_t) \to 0$, and accumulated score estimation errors $\sum_t \epsilon^c(x_t) - \epsilon^u(x_t) \to 0$ along the trajectory are known to correlate with generation quality [3]. Since conditional and unconditional diffusion models share most parameters and differ only in cross-attention layers, larger discrepancies between $\epsilon^c(x_t)$ and $\epsilon^u(x_t)$ present greater learning challenges and potentially induce larger prediction errors. Through examination of the prediction gap along trajectories, we demonstrate that FSG may benefit from more accurate score estimation, which translates to improved generation performance.

Table 13: Comparison of fixed point iterations with increasing inference steps on the SDXL model with NFE = 50, 100, 150 (↑ denotes higher is better, best results under the same NFE are bolded).

| Datasets | | DrawBench [25] | | | | Pick-a-Pic [17] | | | |
| Method | NFE | IR↑ | HPSv2↑ | AES↑ | CLIP↑ | IR↑ | HPSv2↑ | AES↑ | CLIP↑ |
| --- | --- | --- | --- | --- | --- | --- | --- | --- | --- |
| CFG | 50 | 59.02 | 28.73 | **6.07** | 32.29 | 82.14 | 28.46 | **6.73** | 33.53 |
| FSG (ours) | 50 | **82.81** | **29.42** | 6.01 | **32.65** | **98.59** | **28.89** | 6.60 | **34.32** |
| CFG×2 | 100 | 77.71 | 29.36 | 6.06 | 32.44 | 96.06 | 28.84 | 6.64 | 34.13 |
| CFG-100 | 100 | 67.87 | 28.85 | **6.12** | 32.36 | 85.11 | 28.55 | **6.73** | 33.51 |
| FSG (ours) | 100 | **84.12** | **29.54** | 6.02 | **32.76** | **102.82** | **29.05** | 6.66 | **34.30** |
| CFG×3 | 150 | 83.56 | **29.51** | 5.95 | 32.66 | 102.13 | **29.04** | 6.61 | 34.28 |
| CFG-150 | 150 | 29.37 | 28.12 | **6.05** | 32.43 | 62.50 | 27.91 | 6.65 | 32.98 |
| FSG (ours) | 150 | **88.18** | 29.44 | 5.96 | **32.70** | **104.86** | **29.04** | 6.65 | **34.28** |

Table 14: Quantitative results on SD2 model. Dataset: Pick-a-pic; NFE: 50 and 100.

| NFE | 50 | | | | 100 | | | |
| Method | IR↑ | HPSv2↑ | AES↑ | CLIP↑ | IR↑ | HPSv2↑ | AES↑ | CLIP↑ |
| --- | --- | --- | --- | --- | --- | --- | --- | --- |
| CFG | -92.80 | 25.16 | 5.63 | 29.76 | -30.30 | 26.51 | 5.97 | 31.70 |
| CFG ×1/2 | -92.80 | 25.16 | 5.63 | 29.76 | -9.55 | 26.80 | 5.88 | 32.05 |
| CFG++×1/2 | -71.92 | 25.49 | 5.76 | 30.23 | -2.15 | 26.97 | 5.92 | 32.34 |
| Z-sampling | -6.22 | 26.97 | **6.06** | 32.53 | 0.80 | 27.21 | **6.06** | 32.58 |
| Resampling | -17.30 | 26.54 | 5.83 | 32.01 | -10.29 | 26.87 | 5.91 | 31.94 |
| FSG (ours) | **3.30** | **27.21** | 5.93 | **32.59** | **12.94** | **27.37** | 5.97 | **32.77** |

As illustrated in Figure 8, FSG accelerates fixed-point convergence at NFE=50 by integrating long-interval guidance and prioritizing early-stage iterations, accounting for its superior performance under low computational budgets. At higher NFE levels, however, the benefits of long-interval guidance diminish, resulting in comparable convergence speeds across fixed-point operators. Notably, Resampling exhibits slower mid-term convergence due to its stochastic nature, whereas CFG++ achieves greater stability through a smoother strength scheduler.

## D.5 Comparison of Fixed Point Iterations with Increasing Inference Steps

In this section, we compare the performance difference between extending the fixed point iterations and inference steps. As shown in Table 13, allocating inference resources to increase fixed-point iterations yields greater performance improvements compared to increasing the number of inference steps. Gains from additional inference steps saturate rapidly and exhibit instability when the step count is not evenly divisible by $T$. In contrast, extending fixed-point iterations aligns the denoising trajectory closer to the golden path, producing outputs more consistent with human preferences. For instance, at NFE=100, CFG×2 achieves an approximate IR improvement of 10. Our proposed FSG enhances this advantage by optimizing subproblem-solving strategies.

## D.6 Additional Results on Different Models and Samplers

In this section, we present experimental results for the DDPM sampler [14], SD2 [24], and Hunyun-DiT [18] models under NFE = 50 and NFE = 100 settings, complementing the NFE = 150 case discussed in the main text.

As shown in Table 14 and Table 15, FSG maintains superior performance across IR, HPSv2, and CLIP metrics at NFE = 50 and NFE = 100 for both Hunyun-DiT and SD2 models, consistent with the NFE=150 results. With Hunyun-DiT as the base model, FSG improves the IR metric by +11.39

Table 15: Quantitative results on Hunyuan-DiT model. Dataset: Pick-a-pic; NFE: 50 and 100.

| NFE | 50 | | | | 100 | | | |
| Method | IR↑ | HPSv2↑ | AES↑ | CLIP↑ | IR↑ | HPSv2↑ | AES↑ | CLIP↑ |
| --- | --- | --- | --- | --- | --- | --- | --- | --- |
| CFG | 116.82 | 29.09 | 6.59 | 33.00 | 120.50 | 29.12 | **6.74** | 33.20 |
| CFG ×1/2 | 116.82 | 29.09 | 6.59 | 33.00 | 121.05 | 29.15 | 6.54 | 32.96 |
| CFG++×1/2 | 115.63 | 29.03 | 6.63 | 32.97 | 119.71 | 29.19 | 6.64 | 33.14 |
| Z-sampling | 127.82 | 29.21 | **6.69** | **33.56** | 128.45 | 29.37 | 6.71 | 33.40 |
| Resampling | 116.62 | 29.23 | 6.65 | 33.33 | 121.77 | 29.32 | 6.69 | 33.27 |
| FSG (ours) | **128.21** | **29.40** | 6.68 | **33.56** | **129.70** | **29.38** | 6.68 | **33.48** |

Table 16: Quantitative results on DDPM sampler. Dataset: Pick-a-pic; NFE: 50 and 100.

| NFE | 50 | | | | 100 | | | |
| Method | IR↑ | HPSv2↑ | AES↑ | CLIP↑ | IR↑ | HPSv2↑ | AES↑ | CLIP↑ |
| --- | --- | --- | --- | --- | --- | --- | --- | --- |
| CFG | 61.85 | 27.90 | **6.72** | 32.88 | 81.52 | 28.61 | 6.67 | 33.42 |
| CFG ×1/2 | 61.85 | 27.90 | **6.72** | 32.88 | 83.56 | 28.56 | **6.70** | 33.61 |
| CFG++×1/2 | 67.19 | 28.00 | 6.68 | 33.19 | 81.56 | 28.60 | 6.69 | 33.72 |
| Z-sampling | 83.99 | 28.54 | 6.60 | 33.78 | 87.26 | 28.60 | 6.66 | 33.56 |
| Resampling | 73.58 | 28.23 | 6.61 | 33.34 | 84.04 | 28.51 | 6.67 | 33.55 |
| FSG (ours) | **90.95** | **28.80** | 6.67 | **33.85** | **91.82** | **28.66** | 6.63 | **33.81** |

(NfE = 50) and 9.20 (NFE = 100) over the CFG baseline. For the SD2.0 base model, FSG achieves more pronounced improvements in IR, with gains of +96.10 (NFE = 50) and +43.24 (NFE = 100). These results confirm the model-agnostic characteristic of FSG, demonstrating the plug-and-play compatibility of FSG across different diffusion models and inference steps. Notably, weaker base models exhibit greater benefits from the improved convergence properties of FSG.

The DDPM sampler results under NFE = 50 and NFE = 100 are presented in Table 16. Aligning with the main text's NFE=150 findings, FSG outperforms other methods in all metrics. It shows significant quality improvements at NFE = 50 (IR: +29.10; HPSv2: +0.90) and NFE = 100 (IR: +10.30; HPSv2: +0.39), verifying that FSG's fixed-point iteration strategy effectively enhances both generation quality and text alignment for stochastic samplers under varying computational budgets.

These supplementary experiments collectively demonstrate the robust performance of FSG across different base models, samplers, and computational budgets. The model/sampler-agnostic design of FSG and efficient resource allocation consistently improve generation quality and text alignment.

# E    Visualization

## E.1    Additional Visual Results

In this section, we showcase more images generated by different methods. As visualized in Figure 9, FSG demonstrates superior alignment with target prompts compared to baseline methods. For instance:

- In case (a), FSG generates a single blue pizza, whereas other methods produce multiple red pizzas;
- In (b), FSG accurately renders snowboard on the bench;
- In (c), FSG achieves precise text rendering;
- In (d), FSG strictly adheres to the "in nature" requirement.

Notably, FSG mitigates semantic interference common in generative tasks. For example:

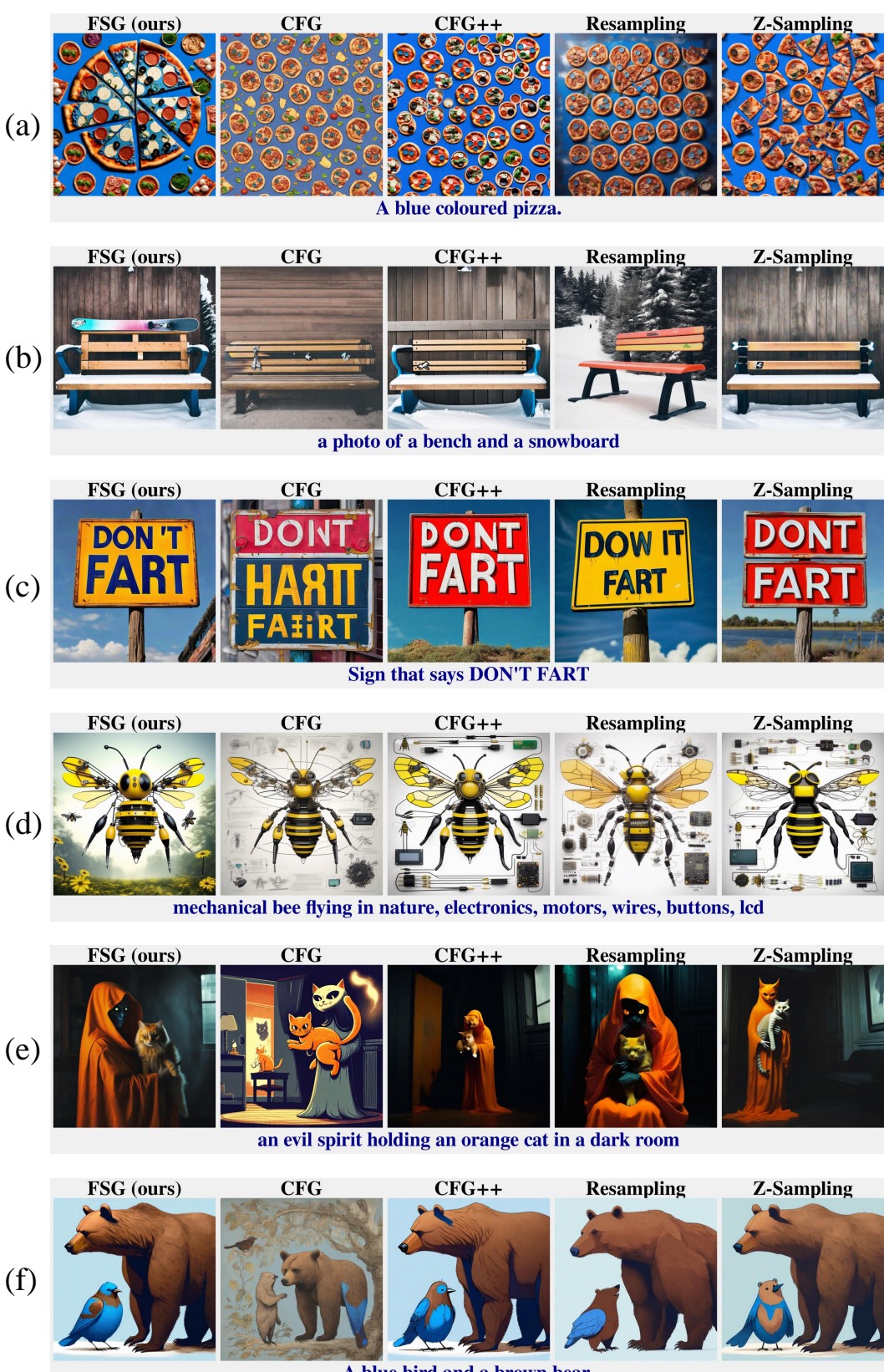

Figure 9: Additional results of qualitative analysis.

A black apple and a
green backpack.

Photo dog animal Vladimir volegov beautiful
blonde freckles woman sensual eating
dining table food cake desert background
glass translucent view Eiffel tower warm
glow neon fireflies fireworks night

A chess queen to the
right of a chess knight

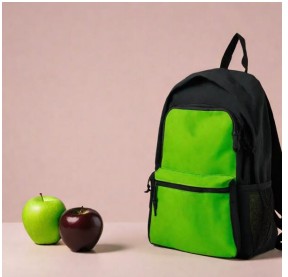
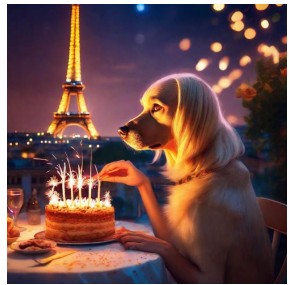
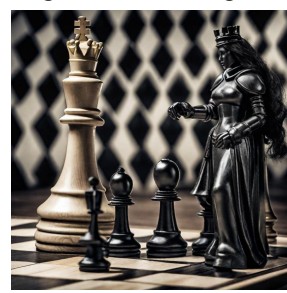

A yellow wall with the
word KA-BOOM on it

A laptop on top of a
teddy bear

A fish eating a pelican.

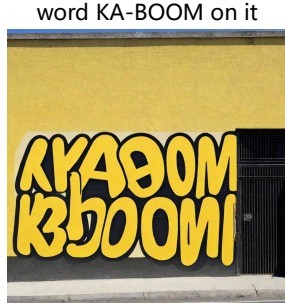
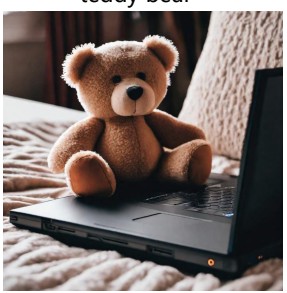
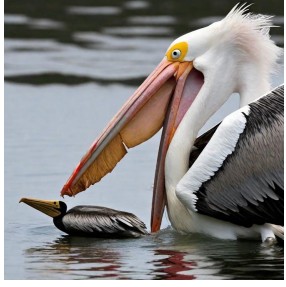

Figure 10: Representative failure cases of FSG.

- In (e), baselines erroneously depict an "evil spirit" as a cat due to the "cat" in prompt, while FSG avoids this bias;

- In (f), baselines incorrectly blend "brown" semantics into a "blue bird" generation, whereas FSG almost preserves color fidelity.

These results highlight the robustness of FSG in aligning with complex prompts, attributed to the long-interval guidance strategy of FSG that stabilizes semantic coherence during generation.

## E.2    Failure Case Analysis

Although FSG demonstrates superior performance across diverse benchmarks, we acknowledge its limitations through failure case analysis. Figure 10 presents representative failure modes encountered by FSG. Most failures originate from inherent deficiencies in the underlying diffusion models. As shown in Figure 10, FSG inherits the base model's difficulties with complex text rendering and counter-intuitive scene compositions.

A notable failure mode specific to FSG stems from its design philosophy of prioritizing strong guidance during early diffusion stages. When the base model exhibits semantic misinterpretations in early timesteps, such as conflating a chess piece with royalty for the term "queen," or overemphasizing individual tokens like "dog", FSG's intensive early-stage calibration can inadvertently reinforce these misconceptions. This amplification effect propagates through subsequent denoising steps, resulting in misaligned generation.

We hypothesize that prompt-aware adaptive guidance, which dynamically adjusts calibration intensity based on semantic understanding, could mitigate this issue. We leave the exploration of such adaptive strategies to future work.

# F   Border Impact and Limitation

**Border Impact.**   The advancement of text-to-image diffusion models carries significant commercial potential for industries such as digital art, advertising, and content creation. A critical challenge in these applications lies in ensuring precise alignment between user prompts and generated outputs, which hinges on effective guidance mechanisms. Our work addresses this challenge by introducing a unified framework of fixed point iterations that reinterprets conditional guidance as a calibration process toward an idealized golden path, thereby decoupling guidance design from sampling dynamics. By demonstrating that prevalent approaches like classifier-free guidance (CFG) and its variants correspond to short-interval iterations within our framework, we reveal their inherent inefficiencies and motivate the proposed foresight guidance (FSG). FSG addresses longer-interval subproblems during early diffusion stages, optimizing computational resource allocation.

The potential impact of our work is threefold. First, it enables practitioners to flexibly integrate diverse guidance mechanisms without theoretical constraints. Second, the framework exhibits extensibility to other domains of conditional generation, such as 3D or video synthesis. Third, it remains compatible with existing techniques like noise search and preference alignment, facilitating test-time scaling (TTS) to enhance image quality by leveraging increased computational resources during inference. Collectively, this work advances adaptive design and offers novel perspectives for unlocking the potential of conditional guidance in diffusion models.

**Limitations**   In this work, we propose a unified framework grounded in fixed point iteration and introduce foresight guidance (FSG) to enhance the alignment and efficiency of conditional guidance. While empirically effective, the framework presents limitations that merit further investigation. First, the concept of the golden path remains empirically observed but not fully theoretically characterized. Second, hyperparameters such as consistency intervals and iteration schedules are determined heuristically rather than through principled optimization. Despite these challenges, our framework provides a foundational step toward systematizing guidance mechanisms, offering a flexible platform for further innovation in efficient and controllable generative modeling.

