# OpenReview forum: "Towards a Golden Classifier-Free Guidance Path via Foresight Fixed Point Iterations"
_NeurIPS.cc/2025/Conference — NeurIPS 2025 spotlight_

### Official Review · Reviewer_Wo6Z · 2025-06-17

**Clarity:** 1
**Significance:** 3
**Originality:** 2
**Rating:** 4
**Confidence:** 4

**Summary:**

This paper proposes improve diffusion CFG sample quality by using fixed point iteration to align unconditional and conditional score outputs during generation. The authors identify four design spaces, consistency interval, fixed point operator, guidance strength, and number of iterations. In Section 3.2, the authors propose Foresight Guidance (FSG), which is a collection of choices for each design space. It is shown that FSG outperforms CFG, CFG++, Z-Sampling, and Resampling on a variety of benchmark tasks.

**Questions:**

I am willing to raise the score to *borderline accept* if the authors provide appropriate explanation for the concerns in **Quality** and **Clarity**.

**Ethical Concerns:**

["NO or VERY MINOR ethics concerns only"]

**Final Justification:**

While the authors' rebuttal has addressed my concerns regarding

- Quality [W1], [W2],
- Clarity [W1], [W2],

there are still remaining concerns with whether $n_{t \rightarrow t+1}$ a valid approximation of $f^{u_{(-1)}}\_{t+1 \rightarrow t}$, and I am still ambivalent about the originality of this work. Hence, I have raised the score to **borderline accept**.

**Limitations:**

The authors discuss the limitations of the work in Appendix D.

**Paper Formatting Concerns:**

None.

**Quality:**

3

**Strengths And Weaknesses:**

Strengths and weaknesses are denoted [S#] and [W#], respectively.

**Quality**
- [S1] Overall, the paper is well-written with clear notations, comprehensive baselines, and thorough ablations. I had no problem understanding the paper.
- [W1] The paper lacks experiments on canonical tasks such as ImageNet generation. I believe results which demonstrate that FSG works on e.g., CIFAR10 or ImageNet class-conditional generation will make the paper much stronger.
- [W2] The paper also lacks diversity metrics such as the Vendi score [1]. I believe the authors should compare quality-diversity trade-off compared to other CFG-based methods to verify whether FSG truly improves the pareto frontier.

**Clarity**

I found some parts of the paper to be written imprecisely, or stated without reference, proof, or analysis. See below.

- [W1] Lines 96 - 98, "when $f^c_{t \rightarrow 0}(x_t)$ and $f^u_{t \rightarrow 0}(x_t)$ differ significantly, the model tends to generate conditional-aligned structures at the expense of texture detail and aesthetic quality. Conversely, when they align, the model balances both conditional alignment and generation quality". I believe supporting this claim with appropriate evidence is crucial for this paper, since this intuition serves as the basis of the proposed method.
- [W2] Lines 99 - 100, "we therefore aim to calibrate $x\_t$ toward $\hat{x}\_t$ by solving $f^u_{t \rightarrow 0}(x_t) = f^c_{t \rightarrow 0}(x_t), x_t \in \mathcal{M}\_t$ where $\mathcal{M}\_t$ denotes the data manifold at time $t$." To my understanding, this paper only solves $f^u\_{t \rightarrow 0}(x_t) = f^c_{t \rightarrow 0}(x_t)$ without the constraint $x_t \in \mathcal{M}\_t$. Indeed, as written at line 251, the authors admit that "excessive iterations risk divergence from the data manifold". I believe that the authors should clarify the proposed method does not solve $f^u\_{t \rightarrow 0}(x_t) = f^c\_{t \rightarrow 0}(x_t), x\_t \in \mathcal{M}\_t$ but solves $f^u_{t \rightarrow 0}(x_t) = f^c_{t \rightarrow 0}(x_t)$.
- [W3] Lines 127 - 128, "approximating $f^{u_{(-1)}}\_{t+1 \rightarrow t}$ by $f^u_{t \rightarrow t+1}$ or $n_{t \rightarrow t+1}$ ...". Is $n_{t \rightarrow t+1}$ a valid approximation of $f^{u_{(-1)}}\_{t+1 \rightarrow t}$? One is stochastic and the other is deterministic.

**Significance**
- [S1] FSG outperforms CFG, CFG++, Z-Sampling, and Resampling on a variety of benchmark tasks on popular models such as SDXL.

**Originality**
- [S1] Interpretation of CFG as a specific instance of FPI is novel and interesting.
- [W1] The proposed method, FSG, is a collection of rather incremental engineering techniques. Specifically, it is a collection of reverse diffusion interval adjustment, FPI, and CFG++ scheduler.

[1] The Vendi Score: A Diversity Evaluation Metric for Machine Learning, TMLR, 2023

---

> ### Author Rebuttal · Authors · 2025-07-31
>
> Thank you for the valuable feedback, here's our response to the concerned problems.
> ___
> **Quality W1/W2: FID, Diversity Metrics, and Class-Conditional Generation**
>
> **A1: FSG improves sample quality and diversity**, as demonstrated by the FID and Vendi scores in Tab. 1 below.
>
> We conducted experiments on the ImageNet 256×256 conditional generation task, generating 1K images per class (totaling 50K images).
> We report FID and Vendi at NFE=25/50. The results show:
> 1. FSG achieves better FID than baselines in class-conditional generation, and even slightly improves diversity.
> 2. CFG/CFG++$\times 2$ (NFE=50) also benefit from increased fixed-point iteration steps compared to CFG/CFG++ (NFE), improving both generation quality and diversity, indicating that fixed-point iterations do not cause mode collapse.
>
> We hypothesize that this is because **fixed-point iterations are performed locally** in the neighborhood of $x_t$, preserving randomness. Intuitively, the initial noise for $x_t$ is $\epsilon$, and fixed-point iteration finds a better $\epsilon'$ in the vicinity of $\epsilon$, enhancing generation quality without sacrificing the inherent randomness of $\epsilon$.
>
> We will provide detailed experimental settings and results in the Appendix.
> **Tab. 1**: FID and Vendi scores for different methods on ImageNet 256×256 conditional generation (lower FID and higher Vendi are better).
> Methods (NFE=25)|FID $\downarrow$ |Vendi $\uparrow$|Methods (NFE=50)|FID $\downarrow$|Vendi $\uparrow$
> -|-|-|-|-|-
> CFG|17.81|3.44|CFG ($\times 2$)|14.69|3.79
> CFG++|13.27|3.91|CFG++ ($\times 2$)|8.85|4.43
> Resample|17.54|3.50|Resample|9.05|4.47
> Z-sampling|19.89|3.40|Z-sampling|8.62|4.64
> FSG|**10.56**|**4.73**|FSG|**7.91**|**5.79**
> ___
> **Clarity W1: Improved Generation Performance When $f^c _ {t\to 0}(x_t)$ Aligns with $f^u _ {t\to 0}(x_t)$**
>
> **A2:** Thank you for your constructive feedback. We provide further evidence to support this statement.
>
> First, we generate data points $x_t$ such that $f^u _ {t\to 0}(x_t) \approx f^c _ {t\to 0}(x_t)$. Specifically, we generate images $x^c$ conditioned on $c$, and obtain $x_t$ via DDIM inversion using unconditional noise $\epsilon^u(x_t)$, i.e., $x_t = f _ {0\to t}^u (x^c)$. These $x_t$ can be regarded as consistent conditionally / unconditionally generation on $c$, i.e., $x^c \approx f^u _ {t\to 0}(x_t) \approx f^c _ {t\to 0}(x_t)$. We then randomly sample a $c' \ne c$ from prompts except $c$, so $x^c \approx f^u _ {t\to 0}(x_t) \ne f^{c'} _ {t\to 0}(x_t)$.
>
> **Tab. 2**: Generation performance of FSG from $x_t$ under matched ($c$) and mismatched ($c' \ne c$) conditions.
>
> $t$|IR (match)|IR (mismatch)|HPSv2 (match)|HPSv2 (mismatch)
> -|-|-|-|-
> 0.8|97.09|-69.96|28.85|25.02
> 0.6|89.45|-171.16|28.73|22.84
> 0.4|84.42|-196.38|28.53|21.86
> 0.2|81.43|-201.60|28.47|21.53
> ___
> It can be seen that when $f^{c/u} _ {t\to 0}(x_t)$ is mismatched, both the alignment and quality of the generated images decrease significantly. Intuitively, the less $f^u _ {t\to 0}(x_t)$ matches the condition $c$, the more components in $x_t$ are unsuitable for condition $c$, making it harder for the diffusion model to correct these components within the $t\to 0$ interval.
>
> We will add a section in the Appendix to introduce the golden path, along with detailed experimental settings and visualization results.
>
> ___
> **Clarity W2: Satisfaction of Manifold Constraints**
>
> **A3:** Here we clarify that satisfying the manifold constraint is meaningful; otherwise, score estimation outside the probability manifold may be inaccurate. In practice, we avoid violating the manifold assumption via empirical settings.
>
> - As shown in Fig. 2(b) (main text), the norm of $\Delta\epsilon (x_t):=\epsilon^c(x_t)-\epsilon^u(x_t)$ in CFG and CFG++ updates is relatively small. Therefore, as long as the guidance strength $w$ is not set too large, $x_t+w\xi_t\Delta\epsilon (x_t)$ can still be regarded as remaining on the probability manifold $P_t(x_t)$.
>
> - For forward/backward operators, we consider that a suitable ODE/SDE sampler $f_{a\to b}$ can sample $x_b\in \mathcal{M}_b$ from $x_a\in \mathcal{M}_a$, thus satisfying the manifold constraint.
>
> Our paper follows these empirical settings and does not focus on strict satisfaction of the manifold constraint. For rigor, we will remove the strong constraint $s.t. x_t \in \mathcal{M}_t$ from the formulas in the main text and instead discuss it in the text.
>
> ___
>
>
> **Clarity W3: $n_{t\to t+1}(x_t)$ as an approximation of $f^u_{t\to t+1}(x_t)$**
>
> **A4:** The use of noise addition as a backward operator was proposed in [1], where the authors argue that $x_t$ is obtained from $x_{t+1}$ via conditional denoising, and that the injected conditional signals are not completely removed by the noise process $n _ {t\to t+1}$. Therefore, $f^{\gamma} _ {t+1 \to t} \circ n_{t\to t+1}(x_t)$ contains more conditional semantics compared to $x_t$.
>
> In our view, $n _ {t\to t+1}$, like $f^u _ {t\to t+1}$, samples $x _ {t+1} \in \mathcal{M}_{t+1}$ from $x_t$ (differing only in whether the sampler is stochastic or deterministic). However, there is still a gap between $n _ {t\to t+1}$ and $f^u _ {t\to t+1}$, which causes Resample requires more iterations to improve the generation quality.
>
> ___
> **Originality W1: FSG as a collection of engineering techniques**
>
> **A5:** Within our fixed-point iteration framework, FSG can be viewed as a composition of settings like the forward-backward operator and the CFG++ scheduler. However, the novelty of our paper lies in proposing a **unified fixed-point iteration framework** that enables the engineering-composable design of guidance algorithms. The effectiveness of the CFG/CFG++$\times N$ instances constructed under our fixed-point iteration framework has also been experimentally validated. We anticipate that the fixed-point iteration framework will provide deeper insights for future guidance design.
> ___
> **Reference**
>
> [1] Repaint: Inpainting using denoising diffusion probabilistic models.

---

> ### Comment · Reviewer_Wo6Z · 2025-08-05
>
> Thank you for the detailed reply! While the authors' rebuttal has addressed my concerns regarding
>
> - Quality [W1], [W2],
> - Clarity [W1], [W2],
>
> there are still remaining concerns with whether $n_{t \rightarrow t+1}$ a valid approximation of $f^{u_{(-1)}}\_{t+1 \rightarrow t}$, and I am still ambivalent about the originality of this work. Hence, I have raised the score to **borderline accept**.

---

> > ### Author Response · Authors · 2025-08-06
> >
> > Thank you for your constructive feedback. We sincerely appreciate your efforts and will carefully incorporate your suggestions into the final version of our paper. Below, we provide further clarification regarding your concerns:
> >
> > **Regarding the Approximation $n _ {t \to t+1}$ for $f^u _ {t \to t+1}$:**
> >
> > We regard $n _ {t \to t+1}$ as a **valid yet suboptimal approximation** of $f^u _ {t \to t+1}$. This stems from its fixed-point objective: Compared to the standard objective $\epsilon^c(x _ t) - \epsilon^u(x _ t) = 0$ (or $f^c _ {t\to t-1}(x _ t) - f^u _ {t\to t-1}(x _ t) = 0$, equivalently) at $x _ t$, the objective of Resample is:
> >
> > $$\epsilon^c(x _ t) - \epsilon^u(x _ t) = \frac{1}{\gamma}\left[a _ t \varepsilon - \epsilon^u(x _ t)\right], $$
> > where $\gamma > 1$ denotes the guidance strength, $\varepsilon$ is Gaussian noise, and $a _ t = \frac{\sqrt{1-\alpha _ t}}{\sqrt{1-\bar{\alpha} _ t} - \sqrt{\alpha _ t-\bar{\alpha} _ t}}$ is a constant.
> > - Theoretically, when $\gamma$ is sufficiently large, objective of Resample aligns with other fixed-point operators.
> > - Practically, however, $\gamma$ is not set extremely large to maintain sample quality. While this results in Resample underperforming to other fixed-point operators, Table 2 in our main text demonstrates that it benefits from increased iteration counts. This empirically confirms its validity as a functional (though suboptimal) choice.
> >
> > Our inclusion of Resample primarily aims to explore the potential of stochastic operators. We will expand the discussion on this aspect in Appendix A.4 (Resample, Unified Framework).
> >
> > **Regarding the Novelty of Our Contribution:**
> >
> > We wish to emphasize that our core innovation lies in being, to the best of our knowledge, the first work to **unify classifier-free guidance (CFG) under a fixed-point iteration perspective.** This unified framework offers key advantages:
> >
> > 1. It decouples guided algorithm design into independent dimensions (e.g., operator scheduler, interval length), enabling systematic comparison of different design choices.
> >
> >  2. It facilitates the redesign and extension of existing algorithms. Within this framework, we have explored novel methods such as FSG and CFG/CFG++ $\times N$, whose effectiveness have been empirically validated.
> >
> > We believe this fixed-point iteration perspective will provide valuable insights for the future development of guidance algorithms.
> >
> > Once again, we sincerely appreciate your time and valuable feedback. We confirm that all your suggestions will be incorporated in the final version.

---

### Official Review · Reviewer_QUUy · 2025-07-02

**Clarity:** 3
**Significance:** 3
**Originality:** 3
**Rating:** 5
**Confidence:** 3

**Summary:**

This paper reframes classifier-free guidance (CFG) for text-to-image diffusion as a fixed-point calibration problem. at each timestep, the latent should lie on a “golden path” where unconditional and conditional denoising trajectories coincide. Existing guidance methods are shown to be single-step, short-sighted fixed-point iterations.  Based on this unified fixed-point view, the authors introduce Foresight Guidance (FSG), which 1- decomposes the diffusion trajectory into fewer, longer intervals, and 2- allocates multiple forward–backward iterations early in the process.  A theoretical bound predicts that such balanced sub-problem sizing minimizes the conditional–unconditional score gap under a fixed compute budget. Empirically, the authors show FSG outperforms several baselines in prompt fidelity, Human Preference Score, ImageReward and Aesthetic score.

**Questions:**

1. What is the exact wall clock time of FSG compared to the baseline in different NFE settings?
2. Can we learn the interval schedule?
3. Can you provide FID values for the experiments? I'm worried about mode collapse.
4. Have you tried empirically verifying the convergence rate in synthetic settings?

Overall I think this paper makes meaningful contributions to controlled generation and will increase my score if the authors provide answers to the questions above.

**Ethical Concerns:**

["NO or VERY MINOR ethics concerns only"]

**Final Justification:**

My main concerns were addressed by authors. I've revised my rating accordingly.

**Limitations:**

Even if diversity decreases (if that’s why FID was omitted) acknowledging this limitation and discussing its impact would significantly improve the paper’s honesty and transparency.

**Quality:**

3

**Strengths And Weaknesses:**

### Strengths:
1. Casting diverse guidance methods as fixed-point iterations. This clarifies design choices.
2.  FSG is derived from a clear theoretical analysis predicting why larger intervals help under limited compute.
3. Extensive evaluation on prompt fidelity and image rewards.
4. FSG outperforms the baselines in almost all considered metrics and settings.

---
### Weaknesses:
1. Image diversity metrics are not reported. The focus of the paper is on alignment and image quality and loss of sample diversity from stronger early calibration is not measured.
2. Wall clock time are not reported. Wall-clock time analysis is paramount to understanding the overhead of extra forward backward passes, and gives reader a clear picture of the paper's practical contributions.
3. (Minor) Several hyper-parameters are introduced which require tuning.
4. (Minor) Qualitative failure cases could be added for when too many iterations are performed, with a discussion on limitations.

---

> ### Author Rebuttal · Authors · 2025-07-31
>
> Thank you for your valuable questions. Please find our responses below.
> ___
> **Q1:  Wall clock time of FSG**
>
> **A1:** Our method (FSG) has **identical wall-clock time** to baselines (Tab. 1).
>
> **Tab. 1**: Running time (seconds per image) for different methods at NFE=50/100/150.
> |NFE|CFG|CFG++|Resample|Z-Sampling|FSG
> |-|-|-|-|-|-
> |50|6.71|6.82|6.48|6.66|6.77
> |100|12.51|12.60|12.49|12.61|12.56
> |150|18.47|18.47|18.26|18.26|18.49
>
> For a fair comparison, all NFEs in the main text count a single backbone call.
> Although FSG employs longer-interval ODE operators, we reduce its computational time by using single-step ODE solver.
> The potential discretization error is acceptable, as fixed-point iterations occur mainly in the early denoising phase.
> As a result, FSG delivers superior performance while maintaining computational efficiency equivalent to the baselines.
> ___
>
> **Q2: Can we learn the interval schedule?**
>
> **A2: Our interval scheduler can be set relatively easily and does not require extensive fine-tuning or learning.**
>
> For example, when NFE=100, we allocate the NFEs in a **3:2:1** ratio across three stages: $1 \rightarrow 0.67$ (early), $0.67 \rightarrow 0.33$ (mid), and $0.33 \rightarrow 0$ (late), with iterations uniformly distributed within each stage.
>
> As shown in Tab. 2, as long as the overall ratio is not violated (e.g., 1:1:1 or 3:2:1), FSG is **robust to hyperparameter choices**: perturbing the intra-stage distribution or operator intervals leads to stable results. We conducted 5 trials with random perturbations to the position of points intra-stage ($\pm 1$) or interval lengths ($\times 70\\%-130\\%$), and observed stable performance.
>
>
> **Tab. 2**: Performance under hyperparameter settings
> Setting|IR|HPSv2|AES|CLIPScore
> -|-|-|-|-
> 3:2:1 (Default)|102.82|29.05|6.66|34.30
> 1:1:1|99.13|28.95|6.67|34.14
> 1:2:3|93.23|28.70|6.63|33.75
> Intra-stage perturbation|102.29 ($\pm$ 1.01)|28.72 ($\pm$ 0.22)|6.59 ($\pm$ 0.04)|34.33 ($\pm$ 0.17)
> Interval perturbation|102.17 ($\pm$ 0.79)|29.04 ($\pm$ 0.02)|6.67 ($\pm$ 0.01)|34.28 ($\pm$ 0.04)
> ___
> **Q3: FID Metric and Mode Collapse**
>
> **A3:** FSG **improves sample quality and diversity**, as demonstrated by the FID and Vendi scores in Tab. 3 below.
>
> Since FID is challenging to use for large-scale text-to-image models, we conducted experiments on the ImageNet 256×256 conditional generation task, generating 1K images per class (totaling 50K images).
> We report FID and Vendi (diversity metric)[1] at NFE=25/50. The results show:
> 1. FSG achieves better FID than baselines in class-conditional generation, and even slightly improves diversity.
> 2. CFG/CFG++$\times 2$ (NFE=50) also benefit from increased fixed-point iteration steps compared to CFG/CFG++ (NFE), improving both generation quality and diversity, indicating that fixed-point iterations do not cause mode collapse.
>
> We hypothesize that this is because **fixed-point iterations are performed locally** in the neighborhood of $x_t$, preserving randomness. Intuitively, the initial noise for $x_t$ is $\epsilon$, and fixed-point iteration finds a better $\epsilon'$ in the vicinity of $\epsilon$, enhancing generation quality without sacrificing the inherent randomness of $\epsilon$.
>
> We will provide detailed experimental settings and results in the Appendix.
>
> **Tab. 3**: FID and Vendi scores for different methods on ImageNet 256×256 conditional generation (lower FID and higher Vendi are better).
> Methods (NFE=25)|FID $\downarrow$ |Vendi $\uparrow$|Methods (NFE=50)|FID $\downarrow$|Vendi $\uparrow$
> -|-|-|-|-|-
> CFG|17.81|3.44|CFG ($\times 2$)|14.69|3.79
> CFG++|13.27|3.91|CFG++ ($\times 2$)|8.85|4.43
> Resample|17.54|3.50|Resample|9.05|4.47
> Z-sampling|19.89|3.40|Z-sampling|8.62|4.64
> FSG|**10.56**|**4.73**|FSG|**7.91**|**5.79**
> ___
> **Q4: Convergence rate in synthetic settings**
>
> **A4: FSG accelerates convergence under the Gaussian mixture setting.**
>
> We simulated the convergence rates of FSG and CFG under a Gaussian mixture setting. Following [2], the target distribution is a Gaussian mixture $\mathcal{N}(\pm \mathbf{m}, \sigma^2I)$ with two equally weighted components ($\mathbf{m}\in \mathbb{R}^d, \lVert \mathbf{m} \rVert_2^2=d$). The conditional sampling target is the component with mean $+\mathbf{m}$. Assuming the score is estimated accurately, [2] shows that CFG achieves faster convergence than using only the conditional prediction $\epsilon^c(x_t)$, as quantified by $\mathbb{E}(x_t)\cdot\mathbf{m}/\sqrt{d}$.
>
> Building on this, we simulated FSG with an interval of 0.1 at $t=0,0.1,...,0.9$, and observed significantly faster convergence. The results indicate that FSG provides stronger and more effective guidance, especially in the early stages.
>
> **Tab. 4**: Convergence speed under the Gaussian mixture setting ($\times 10^{-2}$), using the same metric as in [1] (higher values indicate better performance).
> $t$|$1-e^{-2}$|$1-e^{-1.5}$|$1-e^{-1}$|$1-e^{-0.5}$|0
> -|-|-|-|-|-
> Conditional|0.45|1.09|2.02|4.08|5.77
> CFG ($w=4$)|0.98|2.41|4.12|6.62|11.50
> FSG ($w=4$)|3.47|4.59|6.32|8.89|14.70
>
> We will provide additional details and visualizations of the simulation experiments in the Appendix.
>
> ___
> **Q5: Qualitative failure cases with a discussion on limitations**
>
> **A5:** When excessive iterations are performed, sample quality may degrade due to violation of the manifold constraint $x_t \in \mathcal{M}_t$. We will include qualitative failure cases and discuss these limitations in the Appendix.
> ___
> **Reference**
>
> [1] The Vendi Score: A Diversity Evaluation Metric for Machine Learning
>
> [2] Classifier-Free Guidance: From High-Dimensional Analysis to Generalized Guidance Forms

---

> > ### Comment · Reviewer_QUUy · 2025-08-02
> >
> > I appreciate the authors' response. My main concerns have been addressed.

---

> > > ### Author Response · Authors · 2025-08-02
> > >
> > > We are grateful for your valuable contribution to our work. We’ll make sure to incorporate your suggestions into the final version. Thanks again for your feedback!

---

### Official Review · Reviewer_5edQ · 2025-07-03

**Clarity:** 3
**Significance:** 3
**Originality:** 3
**Rating:** 5
**Confidence:** 4

**Summary:**

The paper proposed a unified framework based on fixed-point iteration that includes variants of classifier-free guidance as special cases, and outlines the design choices under this framework. The author also provides theoretical studies on how to balance the trade-off between the effectiveness and efficiency of iterative denoising image generation. Based on the observation, the author propose the foresight guidance, a new design combination under the framework that decomposes the iterative generation into longer but fewer subproblems. Various ablation and experiments on multiple benchmarks validate the effectiveness of the proposed method.

**Questions:**

1. It would be good to desribe the reasoning behind specific design choices such as the fixed point operaters for the foresight guidance.
2. It would be good to make Table 1 easier to understand.

**Ethical Concerns:**

["NO or VERY MINOR ethics concerns only"]

**Limitations:**

No limitation in the main section.

**Paper Formatting Concerns:**

The reviewer does not have major formatting concerns.

**Quality:**

4

**Strengths And Weaknesses:**

Strength:
1. The paper is generally well-written.
2. The fixed-point iteration view allows clean decomposition of the design spaces for methods based on classifier-free guidance.
3. A theoretically grounded decomposition of iterative denoising generation into subproblems with longer intervals.
4. The experiments are thorough and convincing, investigating multiple controlled scenarios on different benchmarks.

Weakness:
1. Table 1 lacks necessary details in the main text to be fully understood. On top of this, while the proposed method fits into the fixed-point framework, the reasoning behind choosing the specific fixed point operator and scheduler is unclear.

---

> ### Author Rebuttal · Authors · 2025-07-31
>
> Thank you for your valuable suggestions regarding our work. Please find our responses below.
>
> ___
> **Q1: Specific design choices for the foresight guidance: fixed point operator and scheduler.**
>
> **A1:**
>
> 1. In FSG, we select the forward-backward operator with a larger interval $\Delta t$ because it exhibits a **smaller contraction rate**, thereby accelerating the convergence of the fixed-point algorithm.
> We empirically measure local contraction rates for operators $F$ under $\ell_2$ norm: $\hat{r}=\mathbb{E}_{x_0\sim p(x)} \frac{\lVert F(x_t)-F(x_t') \rVert_2^2}{\lVert x_t-x_t' \rVert_2^2}$, where $x_t,x_t'$ are obtained by mixing different noises with $x_0$ (See Tab. 1).
>
> **Tab. 1**:  Empirical contraction rates $\hat{r}$ for different operators across denoising timesteps ($t\in[0,1],dt=0.05$)
> Operators|$t=$ 0.2|0.4|0.6|0.8
> -|-|-|-|-
> $\text{id}-w_t\Delta\epsilon$ (CFG, CFG++)|1.00|1.00|1.00|1.00
>  $f^{\gamma} _ {t+dt\to t}\circ f^{u} _ {t\to t+dt}$ (Z-sampling)|1.04|0.99|0.97|0.99
> $f^{\gamma} _ {t+dt\to t}\circ n^{u} _ {t\to t+dt}$ (Resample)|0.89|0.97|1.03|1.07
> $f^u _ {t/2\to t}\circ f^{\gamma}_{t\to t/2}$|1.03|0.95|0.96|0.98
> $f^u _ {t/4\to t}\circ f^{\gamma} _ {t\to t/4}$|0.61|0.91|0.88|0.91
> $f^u _ {0\to t}\circ f^{\gamma} _ {t\to 0}$|0.62|0.70|0.75|0.79
>
> We observe that, compared to the linear fixed-point operator ($\text{id}-w_t\Delta\epsilon$), the contraction rate of the forward-backward operator decreases as the interval $\Delta t$ increases. Therefore, to balance operator complexity and convergence speed, we choose the forward-backward operator $f^u _ {t - \Delta t \to t}\circ f^{\gamma} _ {t\to t - \Delta t}$ with a larger $\Delta t$.
>
>
> 2. In FSG, we adopt the CFG++ scheduler because it provides **more stable guidance strength** during the early stages.
>
> The schedulers for CFG and CFG++ are given by $\xi_t = \sqrt{1-\bar{\alpha} _ t} - \sqrt{\alpha_t - \bar{\alpha} _ t}$ and $\hat{\xi}_t = \sqrt{1-\bar{\alpha} _ t}$, respectively. Using the $\alpha_t$ setting in DDIM as an example, the derivatives of $\xi_t$ and $\tilde{\xi}_t$ at different timesteps are shown in Tab. 2:
>
> **Tab. 2:** Derivatives of $\xi_t$ and $\tilde{\xi}_t$ at different timesteps
> $t$|0.9|0.8|0.7|0.6|0.5|0.4|0.3|0.2|0.1
> -|-|-|-|-|-|-|-|-|-
> $\xi'_t$ (CFG)|-0.78|-0.71|-0.61|-0.51|-0.41|-0.32|-0.23|-0.09|0.37
> $\tilde{\xi}'_t$ (CFG++)|-0.03|-0.08|-0.15|-0.25|-0.38|-0.51|-0.65|-0.76|-0.93
>
> It can be observed that, in the early stages where the guidance has a greater impact on generation ($t \in [0.67, 1]$), the guidance strength provided by $\xi_t$ decays rapidly, which limits the generation quality.
>
>
> ___
>
> **Q2: Enhancing the comprehensibility of Tab.1 (main text)**
>
> **A2:** Thank you for your valuable suggestions. To improve clarity, we will expand the description of Tab.1 in the main text. This will include complete definitions of all abbreviations in the caption (e.g., $\text{id}$: identity mapping, $\Delta\epsilon(x_t):=\epsilon^c(x_t)-\epsilon^u(x_t)$), as well as clear explanations for "Interval" and "Iter."
>
> Specifically, the Interval $a\to b$ indicates that the method aims to satisfy $f^c_{a\to b} = f^u_{a\to b}$, while Iter describes whether the original version of the method supports increasing the number of fixed-point iterations (Iter=1: not supported; Iter=N: supported). The default versions of CFG, CFG++, and Z-sampling do not mention increasing iterations to improve performance. The author of Resampling noted that iterations can be increased, but did not provide an explanation from the fixed-point perspective.
>
> In contrast, our fixed-point iteration framework naturally enables this enhancement, and we validate the performance of CFG/CFG++ with additional iterations as CFG/CFG++ ($\times 2/3$) in Tab.2 (main text). These clarifications will be incorporated into the main text as well.

---

> > ### Comment · Reviewer_5edQ · 2025-08-05
> > **Thanks for the rebuttal**
> >
> > I appreciate the author's effort. My concern is addressed.

---

> > > ### Author Response · Authors · 2025-08-06
> > >
> > > We sincerely appreciate your insightful feedback. Your suggestions will be thoughtfully integrated into the final manuscript, and we thank you for your valuable contribution to improving this work.

---

### Official Review · Reviewer_eeiK · 2025-07-03

**Clarity:** 2
**Significance:** 3
**Originality:** 4
**Rating:** 4
**Confidence:** 4

**Summary:**

This paper introduces a unified framework for understanding Classifier-Free Guidance (CFG) in text-to-image diffusion models by reframing conditional generation as a fixed-point iteration problem. The objective is to identify a "golden path" where conditional and unconditional generation processes are consistent. The authors demonstrate that existing methods like CFG are special cases of inefficient, single-step "short-sighted" iterations. To address this, they propose Foresight Guidance (FSG), a novel method that solves longer-interval subproblems with multiple iterations, prioritizing the early stages of diffusion. Extensive experiments across multiple datasets and models are presented to show that FSG outperforms state-of-the-art methods in image quality and prompt alignment.

**Questions:**

- **On Computational Overhead:** Could the authors provide a more direct analysis of the computational costs? A comparison of wall-clock time and total UNet evaluations per generated image would be crucial for a complete assessment of the method's practical efficiency.
- **On the Golden Path Proxy:** Figure 2b provides valuable empirical evidence, showing that FSG reduces the single-step _prediction gap_ ($||ε_c(x_t) - ε_u(x_t)||$). However, the paper defines the ideal golden path based on the consistency of the _full remaining trajectory_ ($f^c_{t→0}(x_t) = f^u_{t→0}(x_t)$). Could the authors comment on how well minimizing the single-step prediction gap serves as a proxy for achieving full-path consistency? Is it possible for this proxy to be a poor indicator, for instance, in cases where small, consistent errors in prediction accumulate to cause a large divergence in the final generated images?
- **On the Contraction Assumption:** The theoretical analysis relies on the assumption that the fixed-point operator `F` is a contraction (Assumption 4). Could the authors provide further justification or empirical evidence for this assumption? This appears to be a very strong condition for an operator whose application involves composing two ODE solves, each making repeated calls to a deep neural network.
- **On Hyperparameter Selection:** The appendix notes that the FSG schedule is chosen heuristically. How sensitive is the method's performance to these choices?

**Ethical Concerns:**

["NO or VERY MINOR ethics concerns only"]

**Final Justification:**

After careful consideration of the authors' rebuttal and follow-up responses, I maintain my score of 4: Borderline accept.

**Issues Successfully Resolved:**

- Computational efficiency concerns were addressed with convincing wall-clock timing data showing FSG has equivalent runtime to baselines
- The contraction assumption was supported with empirical evidence across different operators and timesteps
- Hyperparameter sensitivity was demonstrated to be reasonable with robustness analysis

**Areas for Improvement:**

- Algorithm clarity: The core FSG algorithm and its hybrid implementation (selective application at specific timesteps) could be clearer in the main text, as readers currently need to piece together details across sections

- Theoretical consistency: The relationship between the "golden path" motivation (full trajectory consistency), the actual optimization target (interval consistency), and the visualization method (single-step gaps) would benefit from better integration in the main paper
Presentation structure: Some key implementation details in the appendix would enhance reader comprehension if moved to the main text

**Overall Assessment:**

The paper presents solid technical contributions with strong empirical results and novel theoretical insights. The authors have demonstrated good faith in addressing reviewer concerns and provided valuable additional experiments. Importantly, the authors have acknowledged these presentation concerns and committed to specific revisions that would address them, including adding detailed algorithm explanations and relocating key content to the main text.

The technical merit and empirical strength support acceptance, and the authors' commitment to improving clarity suggests the final version will be significantly more accessible.

**Limitations:**

The authors appropriately acknowledge key limitations in Appendix D.

**Quality:**

3

**Strengths And Weaknesses:**

**Strengths:**

- **A Novel and Unifying Conceptual Framework (Originality/Quality):** A key strength of the paper is its novel reframing of conditional guidance. The appendix provides a detailed mathematical unification of several guidance methods (CFG, CFG++, etc.), offering a new theoretical perspective. This presents a useful and systematic framework for analyzing and designing guidance mechanisms.
- **Comprehensive Empirical Validation (Significance):** The proposed FSG method demonstrates strong performance, outperforming baselines across a range of settings. The consistent improvements across these settings suggest the method's robustness and general applicability.
- **Theoretical Grounding (Quality/Originality):** The paper's claims are supported by a theoretical analysis in Appendix B. Theorem 1 provides a motivation, grounded in optimization theory, for why the proposed FSG strategy could be more effective than existing approaches under a finite computational budget.

**Weaknesses:**

- **Insufficient Analysis of Computational Efficiency (Clarity/Significance):** The paper's efficiency claims are based on the Number of Function Evaluations (NFE). This metric is insufficient for a fair comparison, as the computational work encapsulated by a single "function evaluation" differs substantially between FSG and the baselines. For instance, an FSG iteration involves expensive ODE solves, which is not directly comparable to a single UNet pass in CFG. A more direct comparison of wall-clock time or total UNet calls is necessary to substantiate the claims of improved efficiency.
- **High Hyperparameter Complexity (Clarity):** The proposed method introduces considerable hyperparameter complexity. As detailed in Appendix C.1, FSG relies on a manually-designed, heuristic schedule of intervals, interval lengths, and iteration counts. This may hinder practical adoption and reproducibility compared to the relative simplicity of standard CFG.
- **The "Golden Path" Concept Lacks Rigorous Formulation (Significance):** While motivationally effective, the conceptual foundation of the "golden path" remains more metaphorical than rigorously defined. The paper does not fully characterize the properties of this path (e.g., existence, uniqueness), which limits its utility as a formal theoretical tool. (As mentioned in Appendix D)

---

> ### Author Rebuttal · Authors · 2025-07-31
>
> Thank you for the valuable feedback, here's our response to the concerned problems.
> ___
> **Q1: Computational Overhead**
> **A1:** Our method (FSG) has **identical wall-clock time** to baselines (Tab. 1).
>
> **Tab. 1**: Running time (seconds per image) for different methods at NFE=50/100/150.
> |NFE|CFG|CFG++|Resample|Z-Sampling|FSG
> |-|-|-|-|-|-
> |50|6.71|6.82|6.48|6.66|6.77
> |100|12.51|12.60|12.49|12.61|12.56
> |150|18.47|18.47|18.26|18.26|18.49
>
> For a fair comparison, all NFEs in the main text correspond to a single backbone call.
> Although FSG employs longer-interval ODE operators, we reduce its computational time by using single-step ODE solver.
> As fixed-point iterations occur mainly in the early denoising phase, the potential discretization error is acceptable. As a result, FSG delivers superior performance while maintaining computational efficiency equivalent to the baselines.
> ___
> **Q2: Golden Path Proxy**
>
> **A2:** Thank you for your questions, which help improve the clarity of our work. We clarify that while we use $\lVert \epsilon^c(x_t) - \epsilon^u(x_t) \rVert_2^2$ for visualization in Fig. 2 (main text), the **golden path proxy actually minimized in FSG is $f^c _ {t-\Delta t}(x_t) = f^u _ {t-\Delta t}(x_t)$** (see line 180 in the main text).
>
> The rationale for this proxy is as follows: (1) directly targeting $f^c _ {t\to 0}(x_t) = f^u _ {t \to 0}(x_t)$ is suboptimal for optimizing $x_t$, as the long-range generation process causes the dependence of $f^{c/u} _ {t\to 0}(x_t)$ on $x_t$ to diminish. (2) Minimizing $\lVert \epsilon^c(x_t) - \epsilon^u(x_t) \rVert_2^2$ only regularizes the trajectory locally around $x_t$. Therefore, FSG minimizes $f^c _ {t-\Delta t}(x_t) = f^u _ {t-\Delta t}(x_t)$, which balances these two extremes by selecting an appropriate interval length $\Delta t$.
>
> For visualization, we use $\lVert \epsilon^c(x_t) - \epsilon^u(x_t) \rVert_2^2$ because it is equivalent to the **expectation of clean image gap** $\lVert \mathbb{E}^c (x_0 \mid x_t) - \mathbb{E}^u(x_0 \mid x_t) \rVert_2^2$, where $\mathbb{E}^{c/u} (x_0 \mid x_t)$ denotes the conditional/unconditional expectation of the clean image $x_0$ given $x_t$. This expectation approximates $f^{c/u} _ {t\to 0}(x_t)$, with the equivalence following from Tweedie’s formula: $  \mathbb{E}^{c/u} (x_0 \mid x_t) = [x_t - \sqrt{1-\bar{\alpha} _ t} \epsilon^{c/u}(x_t)] / \sqrt{\bar{\alpha} _ t}$.
>
> Empirically, FSG-guided $f_{t\to 0}^{c/u}(x_t)$ also exhibits stronger semantic similarity, indicating improved full-path consistency. To further support this, we will provide additional visualization examples in the Appendix and revise our paper to enhance clarity.
> ___
> **Q3: Contraction Assumption**
>
> **A3:** We empirically measure local contraction rates for operators $F$ under $\ell_2$ norm: $\hat{r}=\mathbb{E}_{x_0\sim p(x)} \frac{\lVert F(x_t)-F(x_t') \rVert_2^2}{\lVert x_t-x_t' \rVert_2^2}$, where $x_t,x_t'$ are obtained by mixing different noises with $x_0$.
>
> **Tab.2** Empirical contraction rates $\hat{r}$ for different operators across denoising timesteps ($t\in[0,1],dt=0.05$)
> Operators|$t=$ 0.2|0.4|0.6|0.8
> -|-|-|-|-
> $\text{id}-w_t\Delta\epsilon$ (CFG, CFG++)|1.00|1.00|1.00|1.00
>  $f^{\gamma} _ {t+dt\to t}\circ f^{u} _ {t\to t+dt}$ (Z-sampling)|1.04|0.99|0.97|0.99
> $f^{\gamma} _ {t+dt\to t}\circ n^{u} _ {t\to t+dt}$ (Resample)|0.89|0.97|1.03|1.07
> $f^u _ {t/2\to t}\circ f^{\gamma}_{t\to t/2}$|1.03|0.95|0.96|0.98
> $f^u _ {t/4\to t}\circ f^{\gamma} _ {t\to t/4}$|0.61|0.91|0.88|0.91
> $f^u _ {0\to t}\circ f^{\gamma} _ {t\to 0}$|0.62|0.70|0.75|0.79
>
> - Notably, long-interval operators (e.g., $f^u _ {0\to t}\circ f^{\gamma} _ {t\to 0}$) demonstrate **lower contraction rates ($\hat{r} < 1$)** compared to short-interval operators such as Z-sampling or Resample. To balance convergence speed and operator complexity, we select an intermediate interval $\Delta t$ ($dt < \Delta t < t$) for the operator $f^u _ {t-\Delta t\to t}\circ f^{\gamma} _ {t\to t-\Delta t}$.
>
> - The contraction rate of an operator depends on the choice of metric. While operators in CFG/CFG++ are non-contracting under the $\ell_2$ norm, they may exhibit contraction in other carefully designed metrics. Empirically, as shown in Tab. 2 (main text), these methods also benefit from additional iterations, supporting the fixed-point framework.
>
> ___
> **Q4: Hyperparameter Selection**
>
> **A4:** Our scheduler assigns more fixed-point iterations to the early denoising stage. Specifically, we distribute NFE in a **3:2:1** ratio across the stages $1 \to 0.67$ (early), $0.67 \to 0.33$ (mid), and $0.33 \to 0$ (late), with iterations uniformly distributed within each stage.
>
> Alternative ratios (such as 1:1:1 or 1:2:3) lead to degraded performance (see Tab. 3). Under the 3:2:1 setting, FSG demonstrates **robustness to hyperparameters**: perturbing the intra-stage distribution or operator intervals yields stable results.
> We conducted 5 trials, randomly perturbing the position of points intra-stage ($\pm 1$) or the interval length ($\times 70\\%-130\\%$) on our chosen hyperparameters.
>
> **Tab. 3**: Performance under different hyperparameter settings
> Setting|IR|HPSv2|AES|CLIPScore
> -|-|-|-|-
> 3:2:1 (Default)|102.82|29.05|6.66|34.30
> 1:1:1|99.13|28.95|6.67|34.14
> 1:2:3|93.23|28.70|6.63|33.75
> Intra-stage perturbation|102.29 ($\pm$ 1.01)|28.72 ($\pm$ 0.22)|6.59 ($\pm$ 0.04)|34.33 ($\pm$ 0.17)
> Interval perturbation|102.17 ($\pm$ 0.79)|29.04 ($\pm$ 0.02)|6.67 ($\pm$ 0.01)|34.28 ($\pm$ 0.04)

---

> > ### Comment · Reviewer_eeiK · 2025-08-06
> >
> > Thanks for the detailed explanations and clarifications. The authors have addressed some concerns, particularly the computational efficiency question with the helpful wall-clock timing data in A1 and the contraction assumption analysis. However, I still think the paper could benefit from clearer presentation in several areas to help readers better understand the method.
> >
> > First, the actual algorithm isn't entirely clear from the main text. I found myself needing to piece together what FSG actually does from details across sections, and it took me quite some re-reading to understand exactly how CFG×2/3 is implemented. For example, the implementation details in the appendix show that FSG is applied only at specific timesteps like {0, 5, 15} while using standard CFG at other timesteps, but this hybrid approach isn't well explained in the main paper.
> >
> > Second, the relationship between FSG and CFG could be clearer. The paper presents FSG and CFG as alternatives, but the implementation is more of a hybrid approach where FSG is used at strategic points. Understanding this hybrid nature is important for grasping the computational costs and the contribution.
> >
> > Third, while A1 addresses the computational efficiency concern with wall-clock times, the NFE accounting still isn't clearly explained in the main text. Moving the key implementation details from the appendix to the main text would help readers understand how the method actually works.
> >
> > Additionally, there remains an inconsistency between the paper's motivation and implementation. Your response A2 actually highlights this more clearly: Figure 1 motivates the "golden path" concept through full trajectory consistency ($f^c\_{t \to 0}(\hat{x}\_t) = f^u\_{t \to 0}(\hat{x}\_t)$), but FSG optimizes for shorter intervals ($f^c\_{t \to t-\Delta t}(x\_t) = f^u\_{t \to t-\Delta t}(x\_t)$), while Figure 2 visualizes single-step gaps ($\|\varepsilon^c(x\_t) - \varepsilon^u(x\_t)\|\_2^2$). The explanation that "directly targeting the full path is suboptimal" is reasonable, but these three different formulations (full trajectory, interval consistency, single-step visualization) could be better connected in the main paper.
> >
> > Lastly, the authors mention noise optimization/selection methods as part of their motivation for the "golden path" hypothesis. Given this connection, it would be interesting to see comparisons to recent noise optimization methods [1,2,3] that also aim to improve generation quality through test-time adjustments.
> >
> > I believe a clear algorithm detailing FSG's hybrid application at specific timesteps would significantly improve clarity. While the technical contributions are solid and the empirical results are strong, I choose to retain my score due to these presentation clarity points.
> >
> >
> > [1] Guo et al. "InitNO: Boosting Text-to-Image Diffusion Models via Initial Noise Optimization". CVPR 2024.
> >
> > [2] Eyring et al. "ReNO: Enhancing One-step Text-to-Image Models through Reward-based Noise Optimization". NeurIPS 2024.
> >
> > [3] Tang et al. "Inference-Time Alignment of Diffusion Models with Direct Noise Optimization". 2024.

---

> > > ### Author Response · Authors · 2025-08-08
> > >
> > > We sincerely appreciate your constructive feedback. We will carefully revise the paper to enhance clarity based on your suggestions. Below are detailed responses to your concerns:
> > >
> > > **Regarding Paper Clarity**
> > >
> > > Due to space limitations, we initially placed some implementation details in the Appendix. Thank you for your advice, we will make the following revisions:
> > > 1. We will add a detailed explanation of FSG implementation in Sec 3.2, including its selective application of FSG operator at specific timesteps while using CFG at others. An annotated algorithm will clarify this structure.
> > > 2. We will relocate the discussion of CFG/CFG++ ($\times N$) obtained through increased fixed-point iterations to Sec 3.1 as part of our contributions.
> > > 3. The experimental section (4.1) will include formal definitions of NFE and report wall-clock time.
> > > 4. "Interval consistency" will be introduced and visualized in the main text to maintain consistency, and discussions of full-trajectory / single-step will be moved to the Appendix.
> > >
> > > **Regarding Comparison with Noise Optimization**
> > >
> > > We clarify that our method is **orthogonal yet complementary to noise optimization**. While noise optimization seeks improved initial points, our FSG (and other guidance algorithms) perform fixed-point calibration during the denoising process. Intuitively, both approaches can be combined to further enhance generation quality. Intuitively, these approaches can be combined for further gains.
> > >
> > > Thank you for highlighting this direction. We explored the comparison and synergy between noise optimization and fixed-point iteration. Due to time constraints, experiments were limited to pretrained Noise Prompt Network (NPNet) [1]. We will expand comparisons in final version. Key observations from Tab. 1 are:
> > >
> > > - FSG demonstrates superior alibration effects during sampling compared to NPNet's noise optimization.
> > >
> > > - FSG synergizes with noise optimization, achieving *"1+1>2"* gains (e.g., 112.64 IR at 50 NFE).
> > >
> > > **Tab. 1:** Synergy between noise optimization and FSG (SDXL, Pick-a-Pic)
> > >
> > > Methods|IR ($\uparrow$)|HPSv2 ($\uparrow$)|AES ($\uparrow$)|CLIP ($\uparrow$)
> > > -|-|-|-|-
> > > Standard (CFG)|82.14|28.46|6.73|33.53
> > > +NPNet|91.66|28.60|6.70|33.57
> > > FSG50|98.59|28.89|6.60|34.32
> > > FSG100|102.82|29.05|6.66|34.30
> > > FSG50+NPNet|112.64|29.04|6.54|34.09
> > > FSG100+NPNet|111.83|29.15|6.57|34.13
> > >
> > > We are profoundly grateful for your time and insightful comments. All suggestions will be incorporated in the final version. We hope this response resolves your concerns and welcome further discussion.
> > >
> > > [1] Golden Noise for Diffusion Models: A Learning Framework.

---

> > > > ### Comment · Reviewer_eeiK · 2025-08-08
> > > >
> > > > Thank you for your thorough responses. Your rebuttal successfully addressed the key technical concerns, particularly with the wall-clock timing data and contraction rate analysis.
> > > >
> > > > I appreciate your commitment to improving the paper's clarity based on my feedback, especially the plans to add detailed algorithm explanations and move key implementation details to the main text. These revisions will significantly enhance the paper's clarity and accessibility.
> > > >
> > > > The work presents solid technical contributions with strong empirical results. I'm maintaining my score of 4, as the technical merit supports acceptance while the presentation improvements you've outlined will make the final version much stronger. I look forward to the revised version.

---

> > > > > ### Author Response · Authors · 2025-08-09
> > > > >
> > > > > We sincerely appreciate your acknowledgment and positive feedback on our work. Your suggestions are invaluable in strengthening the paper, and we will thoughtfully integrate your suggestions into the final manuscript.

---

### Note · Authors · 2025-08-13

**Rebuttal Summary**

We sincerely thank the Area Chair and all reviewers for your time and insightful suggestions. Your feedback has been invaluable in refining our work. We have carefully addressed all points in our response. Below is a summary:

In this work, we present the first unified framework for CFG and its variants through the lens of fixed-point iteration. By redesigning key dimensions (e.g., interval size and iterations), we propose a novel method FSG with experimentally validated effectiveness.

We are encouraged by reviewers’ positive feedback on: unified conceptual framework (Reviewers eeiK, 5edQ, QUUy, Wo6Z), comprehensive experiments & FSG's effectiveness (eeiK, 5edQ, QUUy, Wo6Z), theoretical grounding (eeiK, 5edQ, QUUy) and paper readability (5edQ, Wo6Z).

---

**Additional Experiments & Improvements**

We thank reviewers for constructive suggestions. In response, we conducted additional experiments:

1. **Computational overhead** (eeiK, QUUy): FSG has identical wall-clock time to baselines.
2. **Mode collapse** (QUUy, Wo6Z): Our FID and Vendi scores on ImageNet generation demonstrate that FSG improves generation quality while preserving diversity.
3. **Contraction rate of fixed-Point operators** (eeiK): Operators with longer intervals exhibit significantly smaller contraction rates, validating our design.
4. **Hyperparameter robustness** (eeiK, QUUy): FSG performs stably under mild hyperparameter settings without meticulous tuning.
5. **Convergence in synthetic settings** (QUUy): FSG accelerates early-stage convergence in Gaussian mixture setting.
6. **Comparison with noise optimization** (eeiK): FSG is more effective and exhibits synergistic effects with noise optimization.

---

**Paper Clarity Enhancements**

Per reviewers’ suggestions, we will improve clarity by: 1. relocating implementation details of our method from the Appendix to the main text and adding pseudocode (eeiK); 2. Providing detailed explanations for Table 1 (5edQ); 3. Revising notation for manifold constraints (Wo6Z) and visualization in Fig 2 (eeiK) to prevent confusing; 4. adding discussions on Appendix about motivation for the "golden path" (Wo6Z), reason for FSG's design choice (5edQ), approximation analysis of Resample (Wo6Z) and failure cases (QUUy).

We thank Area Chair and all reviewers again for their time and effort in reviewing our paper and are committed to addressing every issue raised.

---

### Decision · Program_Chairs · 2025-09-17

**Decision:**

Accept (spotlight)

**Comment:**

All reviewers recommend acceptance of this submission, and the AC concurs. The final version should include all
reviewer feedback and suggestions from the discussion.